# FEAT-KD: Learning Concise Representations for Single and Multi-Target Regression via TabNet Knowledge Distillation

**Kei Sen Fong** [1]   **Mehul Motani** [1,2]

## Abstract

In this work, we propose a novel approach that combines the strengths of FEAT and TabNet through knowledge distillation (KD), which we term FEAT-KD. FEAT is an intrinsically interpretable machine learning (ML) algorithm that constructs a weighted linear combination of concisely-represented features discovered via genetic programming optimization, which can often be inefficient. FEAT-KD leverages TabNet's deep-learning-based optimization and feature selection mechanisms instead. FEAT-KD finds a weighted linear combination of concisely-represented, symbolic features that are derived from piece-wise distillation of a trained TabNet model. We analyze FEAT-KD on regression tasks from two perspectives: (i) compared to TabNet, FEAT-KD significantly reduces model complexity while retaining competitive predictive performance, effectively converting a black-box deep learning model into a more interpretable white-box representation, (ii) compared to FEAT, our method consistently outperforms in prediction accuracy, produces more compact models, and reduces the complexity of learned symbolic expressions. In addition, we demonstrate that FEAT-KD easily supports multi-target regression, in which the shared features contribute to the interpretability of the system. Our results suggest that FEAT-KD is a promising direction for interpretable ML, bridging the gap between deep learning's predictive power and the intrinsic transparency of symbolic models.

[1]Department of Electrical and Computer Engineering, National University of Singapore, Singapore. [2]N.1 Institute for Health, Institute for Digital Medicine (WisDM), Institute of Data Science, National University of Singapore, Singapore. Correspondence to: Kei Sen Fong <fongkeisen@u.nus.edu>, Mehul Motani <motani@nus.edu.sg>.

*Proceedings of the $42^{nd}$ International Conference on Machine Learning*, Vancouver, Canada. PMLR 267, 2025. Copyright 2025 by the author(s).

## 1. Introduction

The representation of data is an important factor for the predictive performance of machine learning (ML) algorithms (Bengio et al., 2013) and is a key reason why deep learning-based approaches, which possess large representational capacity are among the top-performing algorithms in many applications (Hinton, 2023). In practice, certain application areas, such as healthcare, demand more than just predictive performance – the explainability of the learned representation becomes a precondition for assessing an ML algorithm's suitability for the problem (Di Martino & Delmastro, 2023). This is why low-complexity ML algorithms are still commonly used in various fields where it is important to understand and explain a model's predictions (Habehh & Gohel, 2021). While some researchers have attempted to create post-hoc explanations to black-box models, these are often not sufficient, and pale in comparison to white-box models with concise representations that provide intrinsic explainability (La Cava et al., 2023; Bordt et al., 2022). In these applications, intrinsic explainability is superior because it provides immediate, transparent insights, fostering accountability in high-stakes domain. This is unlike post-hoc explainability which typically relies on approximations that may misrepresent a model's true decision process. Additionally, intrinsically explainable models are not limited in their ability to leverage post-hoc techniques as well.

FEAT (Feature Engineering Automation Tool) is an ML algorithm that provides a unique white-box model structure with concise representations that provides intrinsic explainability (La Cava et al., 2019). FEAT finds a weighted linear combination of concisely-represented features that are discovered via genetic programming. To illustrate, consider $\hat{y}(\mathbf{x}) = 0.3(x_1 \times x_2) + 0.5(x_3 \div x_4{}^2)$, where $\hat{y}(\mathbf{x})$ is the output and $x_i$ are the features. Here, the two concisely-represented features are $x_1 \times x_2$ and $x_3 \div x_4{}^2$, which has a similar structure to popular indicators, such as the body mass index, *mass $\div$ height$^2$* (Keys et al., 1972). FEAT possesses similar interpretability that linear regression models have, in terms of decomposing the output predictions into independent contributions from features, but allows for more complex features to better fit the dataset. The performance gain more than compensates for the slight decrease in inter-

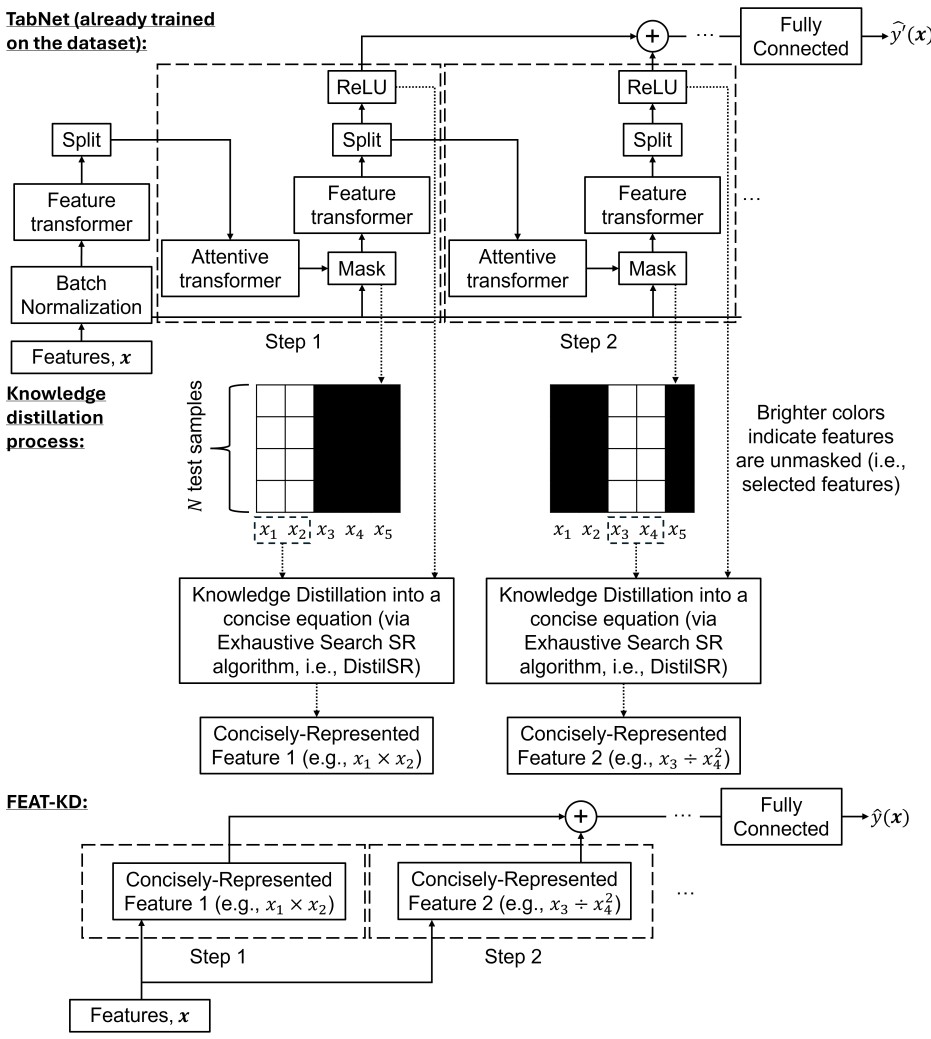

Figure 1: FEAT-KD performs knowledge distillation on an already-trained-TabNet to obtain concisely-represented features. For each 'Step' in TabNet (see dotted boxed 'Step 1', 'Step 2'), the knowledge distillation process takes the top original features selected by the masks in TabNet (e.g., $x_3, x_4$) and uses these subset of original features to discover a concisely-represented feature (e.g., $x_3 \div x_4{}^2$). This is done via a non-genetic-programming exhaustive search symbolic regression algorithm. In notations we will introduce later in Section 2.1, the output of FEAT-KD here is $\hat{y}(\mathbf{x}) = \hat{\beta}_1\phi_1 + \hat{\beta}_2\phi_2$, where $\hat{\beta}_1 = 0.3$, $\phi_1 = x_1 \times x_2$, $\hat{\beta}_2 = 0.5$, $\phi_2 = x_3 \div x_4{}^2$.

pretability compared to vanilla linear regression since the resultant model is still intrinsically explainable (La Cava et al., 2023). Thus, FEAT has become a first-class algorithm for real-world applications that values intrinsic 'white-box' explainability over post-hoc explanations.

To enhance FEAT, we are motivated to couple it with deep-learning-based optimization (such as TabNet) which is well-known and has been successful for a wide range of learning problems. Recent evidence has also shown that genetic programming, which FEAT uses, can be grossly inefficient in symbolic tasks (Kronberger et al., 2024; Fong & Motani,

2023), which motivates research into deep-learning-based optimization. In this work, we propose FEAT-Knowledge Distillation (FEAT-KD), which finds a weighted linear combination of concisely-represented, symbolic features that are derived from piece-wise distillation of a trained TabNet model. In other words, we train TabNet models for regression tasks and perform knowledge distillation on pieces of the TabNet model to convert it into the same model structure that FEAT produces: a weighted linear combination of concisely-represented features (see Figure 1 for a general overview and Table 1 for an example of TabNet's output).

Table 1: **FEAT-KD Example for Multi-Target Regression:** Each target in the regression problem (e.g., atp1d dataset, that has 6 targets) is predicted using a weighted linear combination of the same 6 shared concisely-represented features, $\phi_1$ to $\phi_6$. The weights associated with each feature, $\hat{\beta}_1$ to $\hat{\beta}_6$ are shown here as illustration.

| Concisely-represented feature | | Weight values $\hat{\beta}$ for the $n$-th target: | | | | | |
| --- | --- | --- | --- | --- | --- | --- | --- |
| | | $n = 1$ | $n = 2$ | $n = 3$ | $n = 4$ | $n = 5$ | $n = 6$ |
| $\phi_1 = (x_{297})^{x_{297}} \times x_{119}$ | $\hat{\beta}_1 =$ | 0.00329 | -0.0153 | -0.00938 | -0.00958 | 0.00499 | -0.01 |
| $\phi_2 = (x_{297} + 1.03) \times x_{119}$ | $\hat{\beta}_2 =$ | 0.0378 | -0.0496 | 0.0308 | 0.00413 | 0.0146 | -0.032 |
| $\phi_3 = ((x_{172})^{x_{269}})^{-1.49}$ | $\hat{\beta}_3 =$ | 1.17E-05 | 3.72E-05 | 1.56E-08 | 5.69E-06 | 1.06E-05 | 4.98E-06 |
| $\phi_4 = (-1.33 + x_{269})^{0.0489}$ | $\hat{\beta}_4 =$ | 0.0818 | 0.938 | 0.166 | 0.0231 | 0.238 | -0.0254 |
| $\phi_5 = (x_{182} - 0.809)^{0.106}$ | $\hat{\beta}_5 =$ | 0.0482 | 0.302 | 0.554 | 0.52 | 0.175 | 0.55 |
| $\phi_6 = (0.856 + x_{252}) \times x_{182}$ | $\hat{\beta}_6 =$ | 0.248 | 0.194 | 0.159 | 0.198 | 0.198 | 0.232 |

The main contributions of the paper are as follows:

1. We propose FEAT-Knowledge Distillation (FEAT-KD), which finds the same model structure as FEAT (i.e., a weighted linear combination of concisely-represented, symbolic features), but instead of discovering these features via genetic programming, these features are derived from piecewise knowledge distillation of a trained TabNet model.

2. We demonstrate across a large variety of datasets (inclusive of the regression datasets used in the original TabNet work), that in comparison to TabNet (a deep learning model), FEAT-KD (a symbolic model) reduces TabNet into a concise model which achieves competitive performance.

3. We demonstrate across a large variety of datasets (inclusive of the regression datasets used in the original FEAT work), that FEAT-KD consistently outperforms FEAT and learns models with better prediction performance, smaller size and less complex primitive symbols. In addition, we demonstrate that FEAT-KD easily supports multi-target regression, in which the shared features contribute to the interpretability of the system.

## 2. Related Works

### 2.1. FEAT (Feature Engineering Automation Tool)

FEAT (La Cava et al., 2019) is interested in the task of regression, where the goal is to build a predictive model $\hat{y}(\mathbf{x}) : \mathbb{R}^d \to \mathbb{R}$ using $N$ paired examples $\mathcal{T} = \{(\mathbf{x}_i, y_i)\}_{i=1}^N$, $\mathbf{x} \in \mathbb{R}^d$, $y \in \mathbb{R}$. Note that $\mathbf{x}$ is scaled to zero mean, unit-variance. FEAT uses the fixed form

$$\hat{y}(\mathbf{x}) = \phi(\mathbf{x})^T \hat{\beta}, \qquad (1)$$

where $\phi = [\phi_1, \ldots, \phi_m]^T$ are $m$ concisely-represented features with their respective coefficients $\hat{\beta} = [\hat{\beta}_1, \ldots, \hat{\beta}_m]^T$. Each concisely-represented feature, $\phi_i(\mathbf{x}) : \mathbb{R}^d \to \mathbb{R}$, where $i \in \{1, 2, \ldots, m\}$, is constructed using a fixed predefined primitive symbol set, given in Table 2. FEAT discovers $\hat{y}$ (more specifically, $\phi$) via genetic programming. The genetic programming approach creates a random initial

Table 2: Primitive functions and terminals used to develop the concise representations in FEAT.

| Parameter | Value |
| --- | --- |
| Continuous functions | $+, -, *, /, {}^2, {}^3, \sqrt{}, \sin, \cos, \exp, \log,$ exponent, logit, tanh, gauss, relu |
| Boolean functions | and, or, not, xor, $=, <, \leq, >, \geq$ |
| Terminals | $x_1, x_2 \ldots, x_d$, constants |

population of candidate solutions for $\hat{y}$, which then undergo a selection process, and the selected solutions undergo variations (e.g., crossover, mutation) and the varied solutions compete to survive. This process of selection, variation and survival repeats until a terminal condition is met, most frequently determined by the maximum number of generations of evolution defined by the user.

In FEAT, the selection and survival of candidate solutions for $\hat{y}$ are done via multi-objective selection and survival techniques which considers two key measurements of each candidate solution: 1). mean squared error of $\hat{y}$ against the true labels, $y$, 2). complexity, which is computed by taking a weighted summation of user-defined complexity scores to each primitive symbol (La Cava et al., 2019).

Two other variants of FEAT exist (La Cava et al., 2019). Both aim to increase the disentanglement between the concisely-represented features by adding a third objective.

1. FEAT-Corr, which additionally minimizes the average squared Pearson's correlation among features of $\phi$:

$$\text{Corr}(\phi) = \frac{1}{m(m-1)} \sum_{\phi_i, \phi_j \in \phi, i \neq j} \left( \frac{\text{cov}(\phi_i, \phi_j)}{\sigma(\phi_i) \sigma(\phi_j)} \right)^2.$$

2. FEAT-CN, which additionally minimizes the condition number (CN) of the solutions:

$$CN(\phi) = \frac{\mu_{\max}(\mathbf{\Phi})}{\mu_{\min}(\mathbf{\Phi})},$$

where $\mathbf{\Phi}$ is the $N \times m$ representation matrix and $\mu_{\max}$ and $\mu_{\min}$ are the largest and smallest singular values of $\mathbf{\Phi}$.

In this work, we keep the model structure in (1), but propose to move away from genetic programming and utilize deep-learning-based optimization, which is well-known and has been successful for a wide range of learning problems.

## 2.2. TabNet

TabNet (Arik & Pfister, 2021) is a deep tabular data learning model that can be used for regression. In regression tasks, TabNet demonstrates state-of-the-art performance (Borisov et al., 2022). TabNet uses a sequential attention mechanism that allows the model to focus on different subsets of features at each decision 'Step', mimicking decision trees' while benefiting from neural networks' representational power. By utilizing sparse feature masks that guide the network to attend to specific features, TabNet simultaneously performs feature selection and learning.

In the context of this paper (i.e., knowledge distillation to obtain a model structure as shown in (1)), there are three main reasons for selecting TabNet among other deep learning methods for tabular data:

1. TabNet has a similar model structure as (1): $\hat{y}'(\mathbf{x}) = \varphi(\mathbf{x})^T \hat{\gamma}$, where $\varphi = \left[\varphi_1, \ldots, \varphi_{N_d \times N_{steps}}\right]^T$ are the extracted transformed features with their respective coefficients $\hat{\gamma} = [\hat{\gamma}_1, \ldots, \hat{\gamma}_{N_d \times N_{steps}}]^T$, and $N_d$ is the dimension of the output of the ReLU unit in each 'Step' and $N_{steps}$ are the number of 'Step' (see an example of 'Step 1' and 'Step 2' in Figure 1). If we convert $\varphi$ into concisely-represented symbolic features, we can obtain the same interpretable structure that FEAT has, which we discuss later.
2. TabNet has sparse learnable feature masks, which encourages disentanglement between the transformed features discovered in each 'Step' (see Figure 1). The features selected by the masks in each 'Step' inform us of the main features utilized by the ReLU unit at each 'Step'.
3. TabNet demonstrates the top prediction performance among state-of-the-art deep learning methods for tabular data. (Borisov et al., 2022).

## 2.3. Knowledge Distillation

Knowledge distillation refers to model compression techniques where a smaller student model is trained to replication the behavior of a larger, more complex teacher model (Hinton et al., 2015). In this work, we consider a specific case of knowledge distillation with the objective of compressing pieces of a deep learning model into symbolic concisely-represented features to form a symbolic model. A naive approach would be to collect input-output data from the entire trained deep learning model and run symbolic regression (SR) algorithms, which are a class of algorithms that discover equations from raw data, on this dataset. However, the search space of possible equations is large, thus, SR algorithms perform poorly on such a naive approach.

In prior SR work, there has been success in distilling a graph neural network for interacting particle systems (Cranmer et al., 2020; Lemos et al., 2023). In those works, trained graph neural networks were separated into smaller pieces which are then approximated by concise equations. The approximation is done by collecting input-output data from these pieces, then utilizing SR algorithms to find equations that best fit these data. In our work, we modify the idea and perform knowledge distillation on pieces of TabNet via an exhaustive search of short, and hence interpretable, equations to obtain a knowledge distilled TabNet. To perform the exhaustive search, we utilize DistilSR (Fong & Motani, 2023), an exhaustive search algorithm that demonstrates the best performance for recovering short equations and outperformed state-of-the-art SR algorithms. Note that DistilSR does not use genetic programming or evolutions, it is a deterministic brute-force search with computational cost of approximately $O(d^l)$, where $d$ is the number of unique terminals and symbols, and $l$ is the expression length.

## 2.4. Multi-Target Regression

Multi-target regression (MTR) extends the traditional single-target regression approach by predicting multiple continuous outcomes simultaneously, addressing the limitations of independently modeling each target. Unlike running single-target regression multiple times, MTR leverages the shared features and underlying correlations between targets. This is particularly important in complex real-world applications such as finance (Santana et al., 2019; da Silva et al., 2018), and healthcare (Jain et al., 2024), where the relationships between outputs can help develop more accurate models. By using shared features, MTR not only improves prediction accuracy but also enhances the interpretability of the entire predictive system, since the size of the model can be reduced by using shared features (see Table 1).

## 3. Methodology

FEAT-KD finds a weighted linear combination of concisely-represented, symbolic features that are derived from piecewise distillation of a trained TabNet model. We train TabNet models for regression tasks and perform knowledge distillation on pieces of the TabNet model to convert the entire TabNet into the same model structure that FEAT produces: a weighted linear combination of concisely-represented features. The FEAT-KD algorithm has 5 main phases:

**Phase 1, Training TabNet:** Fit a TabNet regressor to the dataset, that learns a function $\hat{y}'(\mathbf{x}) = \varphi(\mathbf{x})^T \hat{\gamma} : \mathbb{R}^d \to \mathbb{R}$ using $N$ paired examples $\mathcal{T} = \{(\mathbf{x}_i, y_i)\}_{i=1}^N$.

**Phase 2, Extracting TabNet Transformed Features:** From the trained TabNet model, extract $N_d \times N_{steps}$ transformed-features-datasets (each of the $N_{steps}$ 'Step' in Tabnet produces $N_d$-dimension output). We denote these datasets as $\mathcal{T}_j = \{(\mathbf{x}_i, \varphi_j(\mathbf{x}_i))\}_{i=1}^N, \forall j \in N_d \times N_{steps}$.

**Phase 3, Extracting TabNet Masks for Feature Selection:** For each 'Step' in TabNet (see Figure 1), extract the mask matrix and select the 3 features that is masked the least on average (across datapoints), i.e., most significant 3 features. Modify $\mathcal{T}_j, \forall j \in N_d \times N_{steps}$ to only include the top 3 features of the mask that corresponds to the 'Step' that the transformed-features were obtained from.

**Phase 4, Knowledge Distillation of TabNet 'Step':** Train DistilSR (Fong & Motani, 2023), a non-evolutionary-based SR algorithm that performs exhaustive search of equations, on $\mathcal{T}_j, \forall j \in N_d \times N_{steps}$ to generate $N_d \times N_{steps}$ concisely-represented features, $\phi = \left[\phi_1, \ldots, \phi_{N_d \times N_{steps}}\right]^T$. Note that here, though the mean squared error (MSE) of the candidate equation against the transformed-features is reasonable, we chose to use the affine-invariant MSE (AFI-MSE) in anticipation of Phase 5 where we know the distilled features obtained here will undergo a natural affine transformation in the linear regressor. AFI-MSE is a metric designed to measure the discrepancy between two vectors, $y$ and $\hat{y}$, after optimally aligning $\hat{y}$ with $y$ through an affine transformation. Specifically, AFI-MSE minimizes the traditional mean squared error by determining the best possible linear scaling and translation of $\hat{y}$ in the form $a + b \times \hat{y}$, where $a$ and $b$ are computed using least squares optimization to minimize:

$$\text{AFI-MSE}(y, \hat{y}) = \frac{1}{N} \sum_{i=1}^N (y_i - (a + b \times \hat{y}))^2. \quad (2)$$

AFI-MSE is especially suitable for our purpose as it evaluates the similarity of two vectors, providing a more robust and invariant measure of fit. Consider a simple example with true outputs $y = [0, 12, 28]$ and inputs $x_1 = [1, 2, 3]$ and $x_2 = [4, 5, 6]$. A candidate equation $x_1 \times x_2$, produces predictions $[4, 10, 18]$, which under traditional MSE yield a relatively high value that would lead to the equation not being picked. However, when using AFI-MSE, which first optimally scales and shifts the predictions to best match the true values, we can find parameters $a$ and $b$ (specifically, $a = -8, b = 2$) such that the adjusted prediction $-8 + 2 \times (x_1 \times x_2)$ exactly equals $y$, resulting in a zero error. Thus, AFI-MSE recognizes that while the candidate's predictions differ in scale and shifts, their underlying pattern aligns perfectly with the true values, highlighting the candidate's potential as a good fit despite the initial magnitude mismatch. This is suitable particularly for FEAT-KD because in 'Phase 5', this equation will be scaled and shifted anyway because it is being used as a feature in linear regression. This also effectively simplifies the search space.

Table 3: Primitive functions and terminals used to develop concise representations in FEAT-KD.

| Parameter | Value |
|---|---|
| Continuous functions | $+, -, *, /,$ exponent |
| Terminals | $x_1, x_2 \ldots, x_d,$ constants |

Finally, based on recent criticism and insights on explainability and interpretability in SR, we modified the set of primitive functions and terminals to exclude complex symbols that reduces explainability (Petersen et al., 2019) and restricted the length of the equation to be less than or equal to 5 symbols, in-line with the empirically researched limits of the number of cognitive concepts (Trazzi & Yampolskiy, 2020) and symbols (Matricciani et al., 2019) the mind can retain.

**Phase 5, Refitting of Knowledge Distilled Features:** Perform simple linear regression on the concisely-represented features obtained from the previous 'Step' and the output, $\{(\phi(\mathbf{x}_i), y_i)\}_{i=1}^N$, to obtain $\hat{\beta}$ in $\hat{y}(\mathbf{x}) = \phi(\mathbf{x})^T \hat{\beta}$.

Referring to Figure 1, Phase 1 is illustrated by the trained TabNet. Phase 2 is illustrated by extracting the transformed features from the ReLU units in the dotted boxed, e.g., 'Step 1'. Phase 3 is illustrated by the mask obtained from 'Step 1', in which the features $x_1, x_2$ are selected (least masked) from the mask. This means that in the next phase, the knowledge distillation of 'Step 1' will only be done with $x_1, x_2$ exclusively. Note that two features are selected in the diagram, but in our experiments (as stated above), we select the top three features instead. Phase 4 utilizes DistilSR (minimizing the objective in (2)) to produce the equation, $x_1 \times x_2$, via exhaustive search of all possible equations with a maximum of 5 symbols built from $x_1, x_2$ and the other primitive symbols in Table 3. Phase 5 combines all the concisely-represented features through linear regression, to produce the model presented by FEAT-KD.

## 4. Hyperparameter Details and Tuning

In the experiments, to create and distill models which are more interpretable, we chose to distill TabNet models with $N_d = 2, N_{steps} = 3$, which creates a small set of $N_d \times N_{steps} = 6$ concisely-represented features. We obtained these values via preliminary experiments on a small subset of the data to prevent data leakage (e.g., target 1 of atp1d). The criterion for tuning is based on the test set $R^2$ score performance. We also enabled early-stopping, with 1000 max epochs and a patience of 50.

For DistilSR, which was used to convert 'Step' in TabNet into symbolic equations, we considered recent criticism and insights on explainability and interpretability in SR and modified the set of primitive functions (see Table 2)

and terminals to exclude complex symbols that reduces explainability (Petersen et al., 2019). We also restricted the length of the equation to be less than or equal to 5 symbols, in-line with the empirically researched limits of the number of cognitive concepts (Trazzi & Yampolskiy, 2020) and symbols (Matricciani et al., 2019) the mind can retain. Thus, the hyperparameters for DistilSR were to allow for a max length of 5 and primitive symbol set as shown in Table 2.

Table 4: Varying $N_d$, keeping $N_{steps} = 3$.

| $N_d$ | $R^2$ score |
|---|---|
| 2 | **0.652 (0.074)** |
| 3 | 0.637 (0.053) |
| 4 | 0.528 (0.215) |

In Phase 1 of FEAT-KD, by varying $N_d$ and keeping $N_{steps} = 3$, we observed that increasing $N_d$ led to overfitting, which reduced the $R^2$ score on the test set despite increasing the learning capacity of the model (see Table 4).

Table 5: Varying $N_{steps}$, keeping $N_d = 2$.

| $N_{steps}$ | $R^2$ score |
|---|---|
| 1 | 0.619 (0.062) |
| 2 | 0.647 (0.047) |
| 3 | **0.652 (0.074)** |
| 4 | 0.509 (0.19) |

In Phase 1 of FEAT-KD, by varying $N_{steps}$ and keeping $N_d = 2$, we observed that increasing $N_{steps}$ led to overfitting as well, which reduced the $R^2$ score on the test set despite increasing the learning capacity of the model (see Table 5). However, decreasing $N_{steps}$ created models that are too simple and that may not be able to capture the complexity of the dataset as well as models with higher $N_{steps}$.

Table 6: Varying both $N_d, N_{steps}$, while $N_d \times N_{steps} = 6$.

| $N_{steps}$ | $N_d$ | $R^2$ score |
|---|---|---|
| 2 | 3 | 0.634 (0.062) |
| 3 | 2 | **0.652 (0.074)** |
| 6 | 1 | 0.623 (0.079) |

Since we decided on having a total of 6 concisely-represented, symbolic features, we also experimented varying both $N_d, N_{steps}$, keeping $N_d \times N_{steps} = 6$. Ultimately, $N_d = 2, N_{steps} = 3$ still had the best empirical performance on the test set (see Table 6).

## 5. Results and Discussion

To evaluate FEAT-KD, we performed a large variety of experiments taking datasets used in (i) the original TabNet paper (Arik & Pfister, 2021), (ii) the original FEAT paper (La Cava et al., 2019), (iii) multi-target regression benchmarks (Spyromitros-Xioufis et al., 2016).

### 5.1. Datasets and Experiment Details

In this work, we can further group the datasets into two broad categories: single-target regression and multi-target regression. For single-target regression: from the original TabNet paper (Arik & Pfister, 2021), we used 6 datasets, Syn1 to Syn6 (Chen et al., 2018), and the Rossmann store sales dataset (Kaggle, 2019). From the original FEAT paper (La Cava et al., 2019), we used 8 PMLB datasets {bodyfat, cpu_act_197, cpu_act_573, cpu_small, house_8L, houses, pm10, puma8NH} (Romano et al., 2021).

For multi-target regression: from the original TabNet paper (Arik & Pfister, 2021), we used the SARCOS dataset (Vijayakumar & Schaal, 2000). From multi-target regression benchmarks (Spyromitros-Xioufis et al., 2016), we used 5 benchmark datasets {atp1d, enb, oes_10, rf1, scm1d}.

All results were averaged across 100 randomly seeded 60-20-20 train-validation-test splits. We evaluate FEAT-KD (ours), FEAT, FEAT-Corr, FEAT-CN and TabNet, and in addition to the standard deviation, we perform the Wilcoxon signed-rank test (Conover, 1999) with Bonferroni-adjustment to account for the increased probability of observing rare events from multiple hypotheses (Dunn, 1961). The simplicity of the symbolic expressions has been argued to be interpretable via disentanglement (La Cava et al., 2019) and also validated by clinicians in studies which uses FEAT structure (i.e., see (1)) (La Cava et al., 2023). In the experiments, to create and distill models which are more interpretable, we chose to distill TabNet models with $N_d = 2, N_{steps} = 3$, which creates a small set of $N_d \times N_{steps} = 6$ concisely-represented features. Interpretability is measured in proxy by model size, which is supported by works such as those by Lage et al. (2019); Abdul et al. (2020). Specifically, we define the model size to be total number of functions and terminals. For example, $x_1 \times x_2$ has a model size of 3, $x_1/4 + 0.62^{x_1}$ has a model size of 7.

**Additional Details in Appendix.** More information on datasets, evaluation metric, computational resources and timing are provided in the Appendix.

### 5.2. Single-Target Regression

**FEAT-KD has the best average prediction performance among symbolic model evaluated.** We tabulate the results

Table 7: **Single-Target Regression:** $R^2$ score (Standard Deviation [SD] in brackets) of FEAT-KD, FEAT variants and TabNet, averaged across 100 randomly seeded 60-20-20 train-validation-test splits on 6 synthetic datasets from Chen et al. (2018) and the Rossmann store sales dataset (Kaggle, 2019) used in the original TabNet paper (Arik & Pfister, 2021), and 8 PMLB datasets (Romano et al., 2021) used by La Cava et al. (2019). Higher is better, best white-box performance in bold.

| | White-box | | | | Black-box |
|---|---|---|---|---|---|
| **Dataset** | FEAT-KD (Ours) | FEAT | FEAT-Corr | FEAT-CN | TabNet |
| Syn1 | **0.941 (0.012)** | 0.937 (0.0031) | 0.935 (0.0042) | 0.936 (0.0036) | 0.993 (0.0014) |
| Syn2 | **0.953 (0.0037)** | 0.936 (0.0042) | 0.934 (0.0038) | 0.935 (0.0041) | 0.996 (0.0017) |
| Syn3 | **0.949 (0.0063)** | 0.938 (0.0037) | 0.938 (0.0037) | 0.937 (0.0039) | 0.991 (0.0018) |
| Syn4 | **0.939 (0.0089)** | 0.93 (0.0045) | 0.928 (0.0048) | 0.929 (0.0042) | 0.987 (0.0034) |
| Syn5 | **0.951 (0.0036)** | 0.933 (0.0042) | 0.932 (0.0041) | 0.932 (0.0043) | 0.991 (0.0027) |
| Syn6 | **0.943 (0.0095)** | 0.931 (0.0032) | 0.93 (0.0031) | 0.929 (0.0029) | 0.991 (0.0021) |
| Rossmann Store Sales | 0.68 (0.0096) | 0.679 (0.0096) | **0.681 (0.01)** | 0.679 (0.009) | 0.685 (0.0096) |
| bodyfat | **0.987 (0.0058)** | 0.973 (0.023) | 0.965 (0.0032) | 0.974 (0.024) | 0.968 (0.014) |
| cpu_act_197 | **0.976 (0.0014)** | 0.963 (0.004) | 0.962 (0.0039) | 0.961 (0.0038) | 0.982 (0.0013) |
| cpu_act_573 | **0.976 (0.0014)** | 0.963 (0.0041) | 0.955 (0.0032) | 0.961 (0.0034) | 0.982 (0.0013) |
| cpu_small | **0.968 (0.0015)** | 0.956 (0.0032) | 0.955 (0.0032) | 0.955 (0.0028) | 0.973 (0.0014) |
| house_8L | **0.585 (0.01)** | 0.492 (0.019) | 0.463 (0.037) | 0.465 (0.037) | 0.659 (0.01) |
| houses | **0.664 (0.0078)** | 0.579 (0.015) | 0.566 (0.012) | 0.567 (0.013) | 0.793 (0.0047) |
| pm10 | 0.181 (0.026) | 0.215 (0.039) | **0.226 (0.034)** | 0.221 (0.039) | 0.135 (0.043) |
| puma8NH | 0.618 (0.026) | **0.628 (0.038)** | 0.56 (0.046) | 0.585 (0.042) | 0.684 (0.0064) |

Table 8: **Single-Target Regression:** Bonferroni-adjusted $p$-values using a Wilcoxon signed-rank test, with the one-sided alternative hypothesis that the distribution of $R^2$ score outperformance of FEAT-KD over the 3 FEAT variants is stochastically greater than a distribution symmetric about zero. * indicates FEAT-KD outperformance is statistically significant (i.e., $p < 0.05$).

| **Dataset** | **FEAT-KD >FEAT** | **FEAT-KD >FEAT-Corr** | **FEAT-KD >FEAT-CN** |
|---|---|---|---|
| Syn1 | 1.81e-01 | 4.20e-03* | 3.63e-02* |
| Syn2 | 2.66e-15* | 2.66e-15* | 2.66e-15* |
| Syn3 | 6.57e-11* | 1.15e-11* | 3.95e-12* |
| Syn4 | 3.37e-08* | 7.98e-10* | 2.64e-09* |
| Syn5 | 2.66e-15* | 2.66e-15* | 2.66e-15* |
| Syn6 | 2.90e-09* | 1.05e-09* | 3.03e-10* |
| Rossmann | 1.33e-05* | 1.00e+00 | 1.99e-01 |
| bodyfat | 2.97e-03* | 2.81e-05* | 1.54e-02* |
| cpu_act_197 | 2.66e-15* | 2.66e-15* | 2.66e-15* |
| cpu_act_573 | 2.66e-15* | 2.66e-15* | 2.66e-15* |
| cpu_small | 2.66e-15* | 2.66e-15* | 2.66e-15* |
| house_8L | 2.66e-15* | 2.66e-15* | 2.66e-15* |
| houses | 2.66e-15* | 2.66e-15* | 2.66e-15* |
| pm10 | 1.00e+00 | 1.00e+00 | 1.00e+00 |
| puma8NH | 1.00e+00 | 9.51e-11* | 3.35e-06* |

of the various methods in terms of $R^2$ score on the single-target regression datasets in Table 7. Among the algorithms which produce a symbolic model (i.e., models with the form

given in (1)), which are FEAT-KD, FEAT, FEAT-Corr and FEAT-CN, the algorithm which shows the best average $R^2$ score is FEAT-KD, with small, non-overlapping standard deviation on some datasets. Even on the few datasets which FEAT-KD does not top, the performance is within 3% of the best average $R^2$ score, with pm10 being the only exception.

**FEAT-KD has consistent statistically significant outperformance over the other symbolic models evaluated.** Besides evaluating the average performance, we also compute and tabulate (see Table 8) the Bonferroni-adjusted $p$-values from the one-sided Wilcoxon signed-rank test. 38 out of the 45 comparisons are significant, with FEAT-KD producing statistically significant outperformance over all 3 FEAT variants for 11 of the 15 datasets.

**Why does FEAT-KD outperform FEAT in prediction?** The performance of FEAT-KD is close to that of TabNet (see Table 7), tracking the good performance of TabNet. Though numerical parameters in symbolic models can be optimized via gradient based methods, the symbols themselves are difficult to optimize. Genetic programming for symbolic models, which FEAT uses, has no theoretical guarantees to date (the proof for NP-hardness for a simplified SR problem was only done recently (Virgolin & Pissis, 2022)), and has been shown recently to be inefficient in terms of neglecting large search spaces of short, interpretable equations (Kronberger et al., 2024; Biggio et al., 2021). FEAT-KD addresses this function search by using TabNet, which has deep-learning-based optimization with a disentanglement mechanism (i.e., sparse learnable masks) to reduce the prob-

Table 9: **Single-Target Regression:** Model Size (SD in brackets) of the models shown in Table 7. Lower is better. Also tabulated are the Bonferroni-adjusted $p$-values using a Wilcoxon signed-rank test, with the one-sided alternative hypothesis that the distribution of model size difference between FEAT-KD and the 3 FEAT variants is stochastically smaller than a distribution symmetric about zero. * indicates FEAT-KD's smaller size is statistically significant (i.e., $p < 0.05$).

| | Model Size | | | | Bonferroni-adjusted $p$-values | | |
|---|---|---|---|---|---|---|---|
| **Dataset** | FEAT-KD (Ours) | FEAT | FEAT-Corr | FEAT-CN | TabNet | FEAT-KD $<$ FEAT | FEAT-KD $<$ FEAT-Corr | FEAT-KD $<$ FEAT-CN |
| Syn1 | **49.0 (0.0)** | 81.8 (32) | 78.7 (32) | 84.1 (36) | 5.7k | 1.47e-14* | 1.25e-13* | 6.67e-15* |
| Syn2 | **48.9 (0.62)** | 81.1 (30) | 77.6 (30) | 76.9 (28) | 5.8k | 3.56e-15* | 3.13e-13* | 1.25e-13* |
| Syn3 | **49 (0.28)** | 86.4 (35) | 78.3 (34) | 77.8 (33) | 6.0k | 2.00e-14* | 8.31e-12* | 3.59e-12* |
| Syn4 | **49 (0.00)** | 80.3 (29) | 78.9 (24) | 84.2 (31) | 6.1k | 4.04e-15* | 8.23e-16* | 2.03e-15* |
| Syn5 | **49 (0.28)** | 84 (37) | 84.7 (33) | 77.5 (30) | 6.2k | 1.36e-14* | 2.99e-15* | 1.43e-13* |
| Syn6 | **48.9 (0.56)** | 79.5 (33) | 75 (30) | 81.8 (33) | 6.3k | 6.64e-13* | 7.55e-13* | 7.69e-14* |
| Rossmann | **49 (0.00)** | 57.3 (29) | 62.3 (23) | 64.7 (28) | 5.5k | 2.49e-01 | 5.41e-06* | 3.12e-06* |
| bodyfat | **49 (0.00)** | 49.8 (32) | 64.1 (43) | 61.2 (28) | 6.4k | 1.00e+00 | 1.30e-02* | 7.99e-04* |
| cpu_act_197 | **49 (0.28)** | 59.9 (29) | 70.5 (33) | 65.3 (30) | 7.2k | 1.17e-02* | 5.77e-08* | 1.02e-05* |
| cpu_act_573 | **48.9 (0.47)** | 62.1 (30) | 72.5 (30) | 63.1 (29) | 7.2k | 1.46e-03* | 2.03e-10* | 5.69e-05* |
| cpu_small | **49 (0.00)** | 65.3 (29) | 68.6 (28) | 69.3 (28) | 6.2k | 7.46e-06* | 9.38e-09* | 4.18e-09* |
| house_8L | **49 (0.00)** | 82.2 (34) | 72 (40) | 69.7 (41) | 5.7k | 4.47e-13* | 2.69e-06* | 3.24e-05* |
| houses | 48.9 (0.39) | **40.6 (24)** | 41.4 (27) | 42.6 (27) | 5.7k | 1.00e+00 | 1.00e+00 | 1.00e+00 |
| pm10 | **48.9 (0.47)** | 64.4 (33) | 83.2 (45) | 70.4 (29) | 5.6k | 4.94e-05* | 3.17e-10* | 7.44e-10* |
| puma8NH | **49 (0.00)** | 94.6 (31) | 104 (31) | 103 (34) | 5.7k | 2.05e-17* | 6.98e-18* | 7.10e-18* |

lems into smaller equation discovery subtasks that can be tackled via exhaustive search. These knowledge distillation subtasks are easy to fit, with the $R^2$ score being 0.707, 0.795 and 0.663 for synthetic, Rossmann and PMLB datasets respectively.

**FEAT-KD has the lowest average model size among all models evaluated and has statistically significant evidence that it is consistently smaller.** We measure the model size (i.e., total number of functions and terminals in the model when represented in the form of (1)) and tabulate these in Table 9. FEAT-KD is smaller, on average, compared to all other models evaluated, on all but one dataset. Although FEAT, FEAT-Corr and FEAT-CN are unstable and produce models of largely varying sizes across the 100 random seeds, the statistical test reveals that FEAT-KD's smaller size is statistically significant, as seen by the rightmost 3 columns in Table 9. 40 out of the 45 comparisons are significant, with FEAT-KD producing statistically significant smaller models over all other models for 12 of the 15 datasets. Note that TabNet is always of greater size than all symbolic models across all datasets and random seeds.

Additionally, it should be noted that FEAT-KD uses a reduced set of more interpretable primitive functions (see Table 3) (Petersen et al., 2019), compared to Table 2.

**FEAT-KD is frequently Pareto-optimal with respect to FEAT and other SR algorithms in SRBench single-target regression datasets.** Although FEAT-KD uses the specific

Table 10: **Multi-Target Regression:** $R^2$ score (SD in brackets) of FEAT and TabNet on the SARCOS dataset (Vijayakumar & Schaal, 2000) used in TabNet paper, averaged across 100 randomly seeded 60-20-20 train-validation-test splits.

| | White-box | | Black-box |
|---|---|---|---|
| $n$-th target | FEAT-KD (Ours) | FEAT, FEAT-Corr, FEAT-CN | TabNet |
| $n = 1$ | 0.577 (0.061) | N.A. | 0.449 (0.014) |
| $n = 2$ | 0.612 (0.04) | N.A. | 0.788 (0.0068) |
| $n = 3$ | 0.675 (0.042) | N.A. | 0.883 (0.0045) |
| $n = 4$ | 0.887 (0.049) | N.A. | 0.898 (0.0029) |
| $n = 5$ | 0.295 (0.11) | N.A. | 0.279 (0.01) |
| $n = 6$ | 0.264 (0.02) | N.A. | 0.369 (0.014) |
| $n = 7$ | 0.862 (0.053) | N.A. | 0.906 (0.0022) |

form given in (1) for interpretability reasons, like FEAT, it can also be positioned in the broader literature of SR algorithms. We evaluate FEAT-KD on all 88 datasets used by La Cava et al. (2019) against FEAT and other SR algorithms, and FEAT-KD is Pareto-optimal for 62.0% of them with respect to the other SR algorithms. More discussion and results are included in Appendix F.

### 5.3. Multi-Target Regression

**FEAT-KD easily supports multi-target regression.** TabNet supports multi-target regression by modifying the final

Table 11: **Multi-Target Regression:** $R^2$ score (SD in brackets) of FEAT and TabNet on other multi-target regression datasets (Spyromitros-Xioufis et al., 2016), averaged across 100 randomly seeded 60-20-20 train-validation-test splits. FEAT-KD performs competitively with TabNet on average. Among a total of 54 targets (including SARCOS from Table 10), FEAT-KD performs better on 32 of the targets.

| Dataset | $n$-th target | FEAT-KD (Ours) | TabNet |
|---|---|---|---|
| atp1d | $n = 1$ | **0.652 (0.074)** | 0.572 (0.045) |
|  | $n = 2$ | **0.678 (0.064)** | 0.537 (0.064) |
|  | $n = 3$ | **0.679 (0.041)** | 0.526 (0.048) |
|  | $n = 4$ | **0.671 (0.046)** | 0.501 (0.038) |
|  | $n = 5$ | **0.641 (0.067)** | 0.57 (0.054) |
|  | $n = 6$ | **0.646 (0.05)** | 0.474 (0.039) |
| enb | $n = 1$ | **0.85 (0.016)** | 0.434 (0.023) |
|  | $n = 2$ | **0.85 (0.014)** | 0.435 (0.017) |
| oes_10 | $n = 1$ | 0.176 (0.22) | **0.401 (0.13)** |
|  | $n = 2$ | 0.498 (0.033) | **0.609 (0.066)** |
|  | $n = 3$ | 0.378 (0.29) | **0.479 (0.094)** |
|  | $n = 4$ | **0.678 (0.033)** | 0.628 (0.064) |
|  | $n = 5$ | 0.259 (0.21) | **0.386 (0.11)** |
|  | $n = 6$ | **0.731 (0.37)** | 0.622 (0.086) |
|  | $n = 7$ | 0.356 (0.3) | **0.511 (0.082)** |
|  | $n = 8$ | 0.505 (0.29) | **0.508 (0.1)** |
| rf1 | $n = 1$ | 0.853 (0.037) | **0.873 (0.0021)** |
|  | $n = 2$ | **0.178 (0.071)** | 0.051 (0.015) |
|  | $n = 3$ | 0.890 (0.034) | **0.91 (0.0023)** |
|  | $n = 4$ | **0.829 (0.035)** | 0.716 (0.029) |
|  | $n = 5$ | **0.820 (0.041)** | 0.779 (0.034) |
|  | $n = 6$ | **0.540 (0.065)** | 0.445 (0.013) |
|  | $n = 7$ | **0.746 (0.11)** | 0.679 (0.14) |
|  | $n = 8$ | **0.731 (0.087)** | 0.45 (0.02) |
|  | $n = 9$ | 0.816 (0.027) | **0.884 (0.0033)** |
|  | $n = 10$ | **0.172 (0.057)** | 0.036 (0.017) |
|  | $n = 11$ | 0.795 (0.021) | **0.888 (0.0032)** |
|  | $n = 12$ | **0.693 (0.039)** | 0.643 (0.018) |
|  | $n = 13$ | 0.788 (0.043) | **0.795 (0.04)** |
|  | $n = 14$ | **0.494 (0.068)** | 0.427 (0.018) |
|  | $n = 15$ | **0.770 (0.096)** | 0.683 (0.13) |
|  | $n = 16$ | **0.688 (0.078)** | 0.494 (0.026) |
| scm1d | $n = 1$ | **0.882 (0.0033)** | 0.702 (0.011) |
|  | $n = 2$ | **0.874 (0.0042)** | 0.701 (0.015) |
|  | $n = 3$ | **0.849 (0.0057)** | 0.718 (0.0072) |
|  | $n = 4$ | **0.846 (0.0032)** | 0.718 (0.0053) |
|  | $n = 5$ | 0.405 (0.019) | **0.634 (0.013)** |
|  | $n = 6$ | 0.386 (0.016) | **0.612 (0.013)** |
|  | $n = 7$ | 0.426 (0.027) | **0.644 (0.022)** |
|  | $n = 8$ | **0.792 (0.0047)** | 0.632 (0.021) |
|  | $n = 9$ | 0.602 (0.045) | **0.653 (0.01)** |
|  | $n = 10$ | 0.592 (0.038) | **0.654 (0.0094)** |
|  | $n = 11$ | **0.857 (0.003)** | 0.668 (0.0075) |
|  | $n = 12$ | **0.833 (0.0042)** | 0.685 (0.0088) |
|  | $n = 13$ | 0.646 (0.049) | **0.666 (0.0051)** |
|  | $n = 14$ | 0.636 (0.032) | **0.674 (0.0058)** |
|  | $n = 15$ | **0.706 (0.05)** | 0.666 (0.0062) |
|  | $n = 16$ | **0.697 (0.036)** | 0.666 (0.0064) |

layer to accommodate multiple-dimension outputs. Since the architecture before the final layer remains the same, FEAT-KD's 5 phases can still be used on TabNet that is trained for a multi-target regression dataset in Phase 1. This allows FEAT-KD to distill multi-target regression TabNet models. See Table 1 for an example on the model found by FEAT-KD. Note that multi-target regression is not supported by FEAT, FEAT-Corr and FEAT-CN, as shown in Table 10. Though it may be possible to extend the original FEAT to multi-target, that is a novel separate work that requires proposing a new algorithm, beyond the scope of this paper. As a simple baseline, we ran the original FEAT separately and independently for each target, but the discovered models were much less interpretable because they do not share common transformed features across targets due to the independence during training. Additionally, genetic programming, which the original FEAT uses, does not scale as well with dimensions (Kronberger et al., 2024; Biggio et al., 2021).

**FEAT-KD has some regularization effect over TabNet.** We tabulate the results for the other multi-target regression datasets in Table 11. FEAT-KD demonstrates competitive performance to TabNet, performing better on average in 32 out of 54 targets. DistilSR is able to learn an equation for the transformed features in TabNet (Phase 4), with the average $R^2$ score for the knowledge distillation being 0.535, 0.838, 0.691, 0.938, 0.761, 0.578 for SARCOS, atp1d, enb, oes_10, rf1, scm1d datasets respectively. Unlike the single-target regression, where TabNet tends to predict better than any symbolic model, FEAT-KD outperforms TabNet in predictions for 32 (out of 52) targets in the multi-target regression case. On inspection of the training $R^2$ score, TabNet over-fitted to certain targets and did not generalize well to perform well on the test set, whereas FEAT-KD, a simple model, has training score closer to its test score in Table 11.

## 6. Conclusion

In this work, we propose FEAT-KD, an algorithm which finds a weighted linear combination of symbolic, concisely-represented features. FEAT-KD converts TabNet from a 'black-box' model to an intrinsically explainable 'white-box' model, with competitive prediction performance. Compared to FEAT (including FEAT-Corr, FEAT-CN), FEAT-KD uses a completely different model discovery approach: knowledge distillation of pieces of TabNet via exhaustive search SR instead of genetic programming. FEAT-KD demonstrates improvements over FEAT in terms of: (i) prediction performance, (ii) model size, (iii) complexity of primitive symbols, (iv) support for multi-target regression.

## Acknowledgements

This research/project is supported by the National Research Foundation, Singapore under its AI Singapore Programme (AISG Award No: AISG3-PhD-2023-08-052T), and A*STAR, CISCO Systems (USA) Pte. Ltd and National University of Singapore under its Cisco-NUS Accelerated Digital Economy Corporate Laboratory (Award I21001E0002).

## Impact Statement

This paper presents work whose goal is to advance the field of Machine Learning. There are many potential societal consequences of our work, none which we feel must be specifically highlighted here.

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

## A. Dataset Details and Sources

All datasets used are open-sourced and publicly available. We performed a large variety of experiments taking datasets used in (i) the original TabNet paper (Arik & Pfister, 2021), (ii) the original FEAT paper (La Cava et al., 2019), (iii) multi-target regression benchmarks (Spyromitros-Xioufis et al., 2016).

In this work, we can further group the datasets into two broad categories: single-target regression and multi-target regression. For single-target regression: from the original TabNet paper (Arik & Pfister, 2021), we used 6 datasets, Syn1 to Syn6 (Chen et al., 2018), and the Rossmann store sales dataset (Kaggle, 2019). From the original FEAT paper (La Cava et al., 2019), we used 8 PMLB datasets {bodyfat, cpu_act_197, cpu_act_573, cpu_small, house_8L, houses, pm10, puma8NH} (Romano et al., 2021).

For multi-target regression: from the original TabNet paper (Arik & Pfister, 2021), we used the SARCOS dataset (Vijayakumar & Schaal, 2000). From multi-target regression benchmarks (Spyromitros-Xioufis et al., 2016), we used 5 benchmark datasets {atp1d, enb, oes_10, rf1, scm1d}.

All results were averaged across 100 randomly seeded 60-20-20 train-validation-test splits.

We also perform comparison with SRBench in which the dataset names and random seeds are provided in Appendix F. To be consistent with SRBench, the results are evaluated on 25% test set splits on the exact same random seeds.

## B. Additional Evaluation Metric Details

We measure the metric, $R^2$ score, also known as the coefficient of determination, which measures the proportion of the variance in the dependent variable that is predictable from the independent variables. It is defined as:

$$1 - \frac{\|\mathbf{y} - \hat{\mathbf{y}}\|_2^2}{\|\mathbf{y} - \bar{y} \cdot \mathbf{1}\|_2^2},$$

where $\mathbf{y}$ is the vector of true labels, $\hat{\mathbf{y}}$ is vector of the ensemble-combined predictions and $\bar{y}$ is the mean of the true labels. $\|\cdot\|_2^2$ represents the Euclidean norm (or 2-norm).

We measure the metric, model size, which is total number of functions and terminals. For example, $x_1 \times x_2$ has a model size of 3, $x_1/4 + 0.62^{x_1}$ has a model size of 7.

For statistical comparison, we performed the Wilcoxon signed-rank test with Bonferroni-adjustment to account for the increased probability of observing rare events from multiple hypotheses.

For $R^2$ score, we use the one-sided alternative hypothesis that the distribution of $R^2$ score outperformance of FEAT-KD over the 3 FEAT variants is stochastically greater than a distribution symmetric about zero.

For model size, we use the one-sided alternative hypothesis that the distribution of model size difference between FEAT-KD and the 3 FEAT variants is stochastically smaller than a distribution symmetric about zero.

## C. Computational Resources and Timing

The experiments were ran on Intel(R) Xeon(R) CPU E5-2627 v4@2.30GHz with 128GB RAM, on 100 random seeds per experiment using a 60-20-20 train-validation-test split. For all algorithms, each seeded run is given a max walltime of 3600 seconds.

## D. InterpreTabNet Variant of FEAT-KD

Table 12: **Single-Target Regression:** $R^2$ score (SD in brackets) of FEAT-KD distilling TabNet and InterpreTabNet. Higher is better.

| Dataset | FEAT-KD (TabNet) | FEAT-KD (InterpreTabNet) |
|---|---|---|
| bodyfat | 0.987 (0.0058) | 0.990 (0.0051) |
| cpu_act_197 | 0.976 (0.0014) | 0.977 (0.0015) |
| cpu_act_573 | 0.976 (0.0014) | 0.976 (0.0016) |
| cpu_small | 0.968 (0.0015) | 0.966 (0.0015) |
| house_8L | 0.585 (0.01) | 0.588 (0.0096) |
| houses | 0.664 (0.0078) | 0.660 (0.0053) |
| pm10 | 0.181 (0.026) | 0.196 (0.042) |
| puma8NH | 0.618 (0.026) | 0.596 (0.0086) |

InterpreTabNet is a modification of TabNet which has the same structure as TabNet (Si et al., 2024). InterpreTabNet has 2 main contributions, both of which are relevant to FEAT-KD: i). A regularization scheme that maximizes diversity between masks in the TabNet architecture, ii). capturing feature interdependencies by prompting LLMs with the learned masks.

For i)., InterpreTabNet works very similarly to TabNet, just with extra regularization, so we could easily apply FEAT-KD techniques by replacing TabNet with InterpreTabNet in the code implementation, as done in Table 12. The results yield stronger performance on some datasets, but not on all, consistent with the results obtained by Si et al. (2024). Thus, we can say InterpreTabNet allows for a variant of FEAT-KD that performs competitively.

For ii). InterpreTabNet uses LLMs to generate linguistic interpretations of the masks obtained which is not mutually exclusive with FEAT-KD. In fact, InterpreTabNet and FEAT-KD complement each other, in which InterpreTabNet generates a simplified qualitative description, whereas FEAT-KD

Table 13: **Classification:** Accuracy (SD in brackets) of cFEAT-KD, cFEAT variants, TabNet and InterpreTabNet, averaged across 100 randomly seeded 60-20-20 train-validation-test splits on 8 PMLB datasets (Romano et al., 2021) and 3 UCI datasets (Dua & Graff, 2019). Higher is better, best white-box performance in bold.

| Dataset | White-box | | | | | Black-box | |
|---|---|---|---|---|---|---|---|
| | cFEAT-KD (TabNet) | cFEAT-KD (InterpreTabNet) | cFEAT | cFEAT-Corr | cFEAT-CN | TabNet | InterpreTabNet |
| chess | **0.969 (0.0033)** | **0.969 (0.0034)** | 0.949 (0.0055) | 0.943 (0.0042) | 0.945 (0.0036) | 0.985 (0.00086) | 0.987 (0.00086) |
| hypothyroid | **0.963 (0.0024)** | **0.963 (0.0024)** | 0.951 (0.0013) | 0.949 (0.0029) | 0.952 (0.0025) | 0.956 (0.0018) | 0.958 (0.0018) |
| ionosphere | **0.914 (0.014)** | 0.909 (0.011) | 0.885 (0.011) | 0.868 (0.016) | 0.876 (0.012) | 0.873 (0.0061) | 0.871 (0.0057) |
| kr_vs_kp | 0.964 (0.0019) | **0.966 (0.0027)** | 0.950 (0.0062) | 0.945 (0.0068) | 0.945 (0.0035) | 0.983 (0.0022) | 0.986 (0.0028) |
| sonar | **0.815 (0.025)** | 0.806 (0.035) | 0.705 (0.012) | 0.702 (0.030) | 0.746 (0.017) | 0.635 (0.019) | 0.682 (0.036) |
| spambase | 0.934 (0.0024) | **0.935 (0.0024)** | 0.898 (0.0091) | 0.895 (0.0039) | 0.901 (0.0065) | 0.934 (0.0016) | 0.934 (0.0019) |
| spectf | 0.843 (0.018) | **0.845 (0.019)** | 0.812 (0.021) | 0.776 (0.011) | 0.793 (0.017) | 0.836 (0.022) | 0.812 (0.025) |
| tokyo1 | 0.918 (0.0030) | **0.920 (0.0073)** | 0.907 (0.012) | 0.906 (0.0033) | 0.906 (0.0047) | 0.908 (0.010) | 0.914 (0.0052) |
| Diabetes | 0.765 (0.0099) | **0.770 (0.014)** | 0.738 (0.0079) | 0.733 (0.0073) | 0.737 (0.015) | 0.775 (0.015) | 0.777 (0.012) |
| Forest Cover Type | **0.755 (0.0017)** | 0.753 (0.0017) | 0.748 (0.0021) | 0.748 (0.0012) | 0.749 (0.0010) | 0.753 (0.0022) | 0.754 (0.0022) |
| Poker Hand | **0.621 (0.0034)** | **0.621 (0.0026)** | 0.614 (0.0016) | 0.613 (0.0021) | 0.613 (0.0032) | 0.618 (0.0019) | 0.620 (0.0030) |

generates a simplified quantitative description. Thus, from the trained mask, 2 interpretations can be generated, one from using an LLM to generate a linguistic interpretation and the other from the learned equation from FEAT-KD. Thus, the strengths of InterpreTabNet can be subsumed into FEAT-KD by replacing TabNet with InterpreTabNet.

# E. Extension to Classification

One possible way FEAT-KD can be adapted for classification is by using logistic regression in Phase 5 instead of linear regression. Rather than using $\hat{y}(\mathbf{x}) = \phi(\mathbf{x})^T \hat{\beta}$, as shown in (1) for regression, we use a logistic regression model for classification. That is, for an input $\mathbf{x}$ and $K$ classes, the model is defined as $P(y = k \mid \mathbf{x}) = \exp(\phi(\mathbf{x})^T \hat{\beta}_k)/z$, where $z = \sum_{j=1}^{K} \exp(\phi(\mathbf{x})^T \hat{\beta}_j)$, $\phi(\mathbf{x})$ is the feature vector, and $\hat{\beta}_k$ is the coefficient vector for class $k$.

The predicted class is given by

$$\hat{y} = \arg\max_k \ P(y = k \mid \mathbf{x}).$$

We denote the methods with the prefix 'c' to differentiate them from the regression case. To evaluate the classification performance, we replaced $R^2$ score with both accuracy and F1 score instead. The results on accuracy and F1 score are presented in Tables 13 and 14 respectively.

# F. Evaluating FEAT-KD against SR Algorithms

Although FEAT-KD uses the specific form given in (1) for interpretability reasons, it can also be counted as a type of SR algorithm and be positioned in the broader literature of SR algorithms. Using the results from

SRBench, across the 88 datasets in PMLB used in FEAT, i.e., (ESL, SWD, LEV, ERA, USCrime, FacultySalaries, vineyard, auto_price, cpu_act_197, autoPrice, cloud, puma8NH, cpu_small, elusage, pwLinear, machine_cpu, satellite_image, analcatdata_vehicle, wind, vinnie, pm10, analcatdata_neavote, analcatdata_election2000, pollen, pollution, no2, analcatdata_apnea2, analcatdata_apnea1, bodyfat, cpu, cpu_small_562, cpu_act_573, fri_c0_250_5, fri_c3_500_25, fri_c1_500_25, fri_c4_500_25, fri_c3_1000_25, fri_c2_1000_25, fri_c1_100_10, fri_c4_1000_25, fri_c1_1000_10, fri_c2_100_5, fri_c0_1000_10, fri_c2_250_5, fri_c2_500_5, fri_c2_1000_5, fri_c1_250_5, fri_c3_250_10, fri_c0_250_50, fri_c4_500_10, fri_c2_250_25, fri_c2_1000_10, fri_c4_1000_50, fri_c3_1000_10, fri_c0_1000_5, fri_c3_100_5, fri_c1_1000_5, fri_c3_250_5, fri_c4_250_10, fri_c4_500_50, fri_c3_500_5, fri_c3_1000_50, fri_c0_100_10, fri_c2_1000_50, fri_c4_1000_10, fri_c0_100_5, fri_c2_500_5, fri_c2_500_10, fri_c3_1000_5, fri_c1_500_5, fri_c0_500_25, fri_c2_100_10, fri_c0_250_10, fri_c1_500_50, fri_c1_500_10, fri_c2_500_25, fri_c4_250_25, fri_c3_500_50, fri_c3_500_10, fri_c1_250_10, fri_c1_250_50, fri_c0_500_5, fri_c0_500_50, fri_c0_100_25, fri_c0_250_25, fri_c0_500_10, fri_c1_100_5, fri_c2_250_10) and 10 random seeds, i.e., (11284, 11964, 15795, 21575, 22118, 23654, 29802, 5390, 6265, 860), FEAT-KD is Pareto-optimal for 62.0% of them with respect to the other SR algorithms. The rate at which the other SR algorithms DSR, GP-GOMEA, Operon, gplearn, AFP, AFP_FE, AIFeynman, FEAT, EPLEX, ITEA, SBP-GP, BSR, MRGP, FFX, are 83.6%, 75.9%, 52.6%, 48.6%, 25.5%, 25.8%, 8.3%, 23.0%, 23.2%, 4.9%, 8.8%, 15.2%, 1.5%, 2.7%, respectively. In SRBench, Pareto-optimal means that FEAT-KD has the optimal trade-off with prediction performance and equation size, in which there are no other SR algorithms with smaller equation size with better prediction performance. Of note are only 2

Table 14: **Classification:** F1 score (SD in brackets) of cFEAT-KD, cFEAT variants, TabNet and InterpreTabNet, averaged across 100 randomly seeded 60-20-20 train-validation-test splits on 8 PMLB datasets (Romano et al., 2021) and 3 UCI datasets (Dua & Graff, 2019). Higher is better, best white-box performance in bold.

| | White-box | | | | | Black-box | |
|---|---|---|---|---|---|---|---|
| **Dataset** | cFEAT-KD (TabNet) | cFEAT-KD (InterpreTabNet) | cFEAT | cFEAT-Corr | cFEAT-CN | TabNet | InterpreTabNet |
| chess | **0.970 (0.0041)** | 0.969 (0.0034) | 0.952 (0.0051) | 0.947 (0.0043) | 0.948 (0.0039) | 0.986 (0.00092) | 0.987 (0.0015) |
| hypothyroid | 0.955 (0.0036) | **0.956 (0.0028)** | 0.945 (0.0015) | 0.944 (0.0016) | 0.946 (0.0013) | 0.978 (0.00094) | 0.978 (0.0016) |
| ionosphere | **0.912 (0.015)** | 0.908 (0.011) | 0.905 (0.011) | 0.898 (0.016) | 0.894 (0.015) | 0.905 (0.0052) | 0.902 (0.0035) |
| kr_vs_kp | 0.964 (0.0019) | **0.965 (0.0031)** | 0.952 (0.0056) | 0.948 (0.0058) | 0.948 (0.0051) | 0.984 (0.0022) | 0.986 (0.0028) |
| sonar | **0.814 (0.025)** | 0.805 (0.035) | 0.678 (0.036) | 0.702 (0.040) | 0.721 (0.049) | 0.591 (0.027) | 0.623 (0.040) |
| spambase | 0.934 (0.0025) | **0.935 (0.0025)** | 0.865 (0.012) | 0.866 (0.0059) | 0.869 (0.0092) | 0.918 (0.0031) | 0.916 (0.0035) |
| spectf | 0.836 (0.024) | 0.844 (0.021) | **0.863 (0.019)** | 0.850 (0.013) | 0.856 (0.017) | 0.887 (0.021) | 0.876 (0.020) |
| tokyo1 | 0.919 (0.0054) | 0.920 (0.0073) | **0.929 (0.0095)** | 0.926 (0.0066) | 0.927 (0.0054) | 0.930 (0.0085) | 0.935 (0.0080) |
| Diabetes | 0.728 (0.012) | **0.734 (0.017)** | 0.699 (0.0088) | 0.693 (0.0094) | 0.697 (0.016) | 0.740 (0.019) | 0.742 (0.014) |
| Forest Cover Type | **0.554 (0.0041)** | 0.550 (0.0039) | 0.538 (0.0069) | 0.536 (0.0045) | 0.539 (0.0043) | 0.552 (0.0052) | 0.554 (0.0058) |
| Poker Hand | 0.124 (0.0037) | **0.125 (0.0047)** | 0.120 (0.00060) | 0.120 (0.00070) | 0.120 (0.0010) | 0.122 (0.00090) | 0.123 (0.0035) |

Table 15: Comparison of FEAT-KD with FEAT and other SR algorithms using SRBench data aggregated across 88 datasets and 10 random seeds each.

| SR Algorithm | model size | $R^2$ | training time (s) |
|---|---|---|---|
| AFP | 35.4 | 0.688 | 2790 |
| AFP_FE | 36.0 | 0.700 | 2850 |
| AIFeynman | 2240 | -4.03 | 82900 |
| BSR | 19.8 | 0.245 | 13100 |
| DSR | 8.88 | 0.598 | 35100 |
| EPLEX | 56.3 | 0.814 | 7530 |
| FEAT | 79.1 | 0.847 | 6750 |
| FEAT-KD (Ours) | 49.0 | 0.851 | 1550 |
| FFX | 1630 | 0.00612 | 210 |
| GP-GOMEA | 25.1 | 0.794 | 7610 |
| ITEA | 107 | 0.663 | 6240 |
| MRGP | 10800 | 0.521 | 201000 |
| Operon | 63.7 | 0.853 | 1350 |
| SBP-GP | 693 | 0.851 | 155000 |
| gplearn | 17.6 | 0.563 | 23300 |

## G. Ablation of DistilSR

As an ablation study, we substituted DistilSR with Operon (Burlacu et al., 2020), PySR (Cranmer, 2023), DSR (Petersen et al., 2019) and ParFam (Scholl et al., 2025), and tested on the 88 datasets used in FEAT. Like in SR benchmarking, there is a trade-off between model size and prediction performance. The frequency of being on the Pareto-front for FEAT-KD is 88.6%. Whereas substitution with DSR, PySR, ParFam and Operon yielded 54.4%, 25.3%, 19.0% and 1.3% frequency of being Pareto optimal, respectively. We found that DistilSR in FEAT-KD is best able to exploit the expressivity of the search space of short equations to better model the learned representations by TabNet without overfitting. The results also show that the task of fitting the learned representations is different from normal regression and that SR algorithms that tend to generate shorter expressions perform better. Finally, an additional strength of using DistilSR over other algorithms is that the variance is the lowest because the equation structure search space is the same for every run.

algorithms which appear more frequently than FEAT-KD on the Pareto-optimal front: DSR, GP-GOMEA, of which FEAT-KD has a better $R^2$ than DSR in 83.8% of the experiments and a better $R^2$ than GP-GOMEA in 70.7% of experiments. Including Operon which appears less frequently that FEAT-KD on the Pareto-optimal front but is the 4th most frequent, in terms of $R^2$, the trend is Operon>FEAT-KD>GP-GOMEA>DSR and in terms of model size, the trend is DSR<GP-GOMEA<FEAT-KD<Operon. Thus, FEAT-KD is frequently Pareto-optimal, providing a new point on the optimal Pareto front with a unique trade-off between $R^2$ and model size. It is also noteworthy that FEAT-KD appears much more frequently on the Pareto-optimal front than FEAT (62.0% vs 23.0%). We also report the aggregated performance in Table 15.

