# OpenReview forum: "FEAT-KD: Learning Concise Representations for Single and Multi-Target Regression via TabNet Knowledge Distillation"
_ICML.cc/2025/Conference — ICML 2025 poster_

### Official Review · Reviewer_9uvT · 2025-03-10

**Overall Recommendation:** 3

**Summary:**

The paper substitutes the feature search of the SR method FEAT by using the mask of a TabNet to select the important input variable and learn the feature using DistilSR, an exhaustive search method.

**Claims And Evidence:**

The reviewer appreciates that the paper shows the improvement with respect to FEAT (and variants) and that the loss in performance comapred to TabNet is moderate, especially in the multi-task setting. However, for a fair and thorough comparison it is important to use the whole datasets, especially from the FEAT paper.

While the most interesting comparison is definitely with respect to FEAT, it is also important to compare to other SR methods, with results readily available for the standard SR benchmark SRBench.

**Essential References Not Discussed:**

NA

**Experimental Designs Or Analyses:**

The experiments itself are well designed, however, they are missing datasets and baselines (see Claims and Evidence).

An interesting ablation study would be to substitute DistilSR by a different SR algorithm to see if this idea is also able to improve SR in general. It would be beneficial to use a state-of-the-art algorithm (e.g., Operon [1], PySR [2], uDSR [3], RILS-ROLS [4], or ParFam [5]; one of these is enough, or a similar performing algorithm)  for this, since it would also directly show if this method helps to push the field of SR itself.

[1] Kommenda, M., Burlacu, B., Kronberger, G. and Affenzeller, M., 2020. Parameter identification for symbolic regression using nonlinear least squares. _Genetic Programming and Evolvable Machines_, _21_(3), pp.471-501.

[2] Cranmer, M., 2023. Interpretable machine learning for science with PySR and SymbolicRegression. jl. _arXiv preprint arXiv:2305.01582_.

[3] Landajuela, M., Lee, C.S., Yang, J., Glatt, R., Santiago, C.P., Aravena, I., Mundhenk, T., Mulcahy, G. and Petersen, B.K., 2022. A unified framework for deep symbolic regression. _Advances in Neural Information Processing Systems_, _35_, pp.33985-33998.

[4] Kartelj, A. and Djukanović, M., 2023. RILS-ROLS: robust symbolic regression via iterated local search and ordinary least squares. Journal of Big Data, 10(1), p.71.

[5] Scholl, P., Bieker, K., Hauger, H. and Kutyniok, G., 2025. ParFam--(Neural Guided) Symbolic Regression via Continuous Global Optimization. In _The Thirteenth International Conference on Learning Representations_.

**Methods And Evaluation Criteria:**

Do proposed methods and/or evaluation criteria (e.g., benchmark datasets) make sense for the problem or application at hand?

**Other Comments Or Suggestions:**

- While I agree, that TabNet is not the competitor FEAT-KD has to be beat, Table 4 is slightly misleading as TabNet is included but not bold, even though it wins for most data sets.
- The experiments for multi-task would benefit from a SR baseline. Possibly the authors could simply divide the compute budget by the number of dimensions and run standard FEAT separately. This should help to show the performance gain by FEAT-KD.

**Other Strengths And Weaknesses:**

Strengths:
- The method is very well motivated and shows that the idea worked.
- I am sure that the idea itself is useful for the SR community, as partitioning a formula into smaller parts is an extremely usefull tool for SR. This is another reason for my interest in the ablation study.
- The paper is written clearly.
- The extension to multi-task is appreciated.
Weaknesses:
- See above
- By relying on exhaustive search, it is hard to include more base functions like sin, exp, etc., which is why currently this method can only learn rational functions and potences.

**Questions For Authors:**

- Which base functions was FEAT allowed to use?
- Is the TabNet itself also computed on a single CPU? And it counts into the time budget?
- Why is the winner not bold in Table 7.

**Relation To Broader Scientific Literature:**

The paper is well embedded in the literature.

**Theoretical Claims:**

NA

---

> ### Author Rebuttal · Authors · 2025-04-01
>
> > General
>
> Please see response R1 to reviewer mwAW.
> > Claims And Evidence
>
> R19: We thank the reviewer and will add the results for extended datasets in PMLB used in FEAT in Appendix F, “Evaluation on More Datasets”. We pick a representative subset in the main pages because we want to show results on datasets from a diverse range of independent sources (synthetic and real-world data from both TabNet and FEAT, and multi-target regression datasets).
>
> R20: We thank the reviewer for the pointer. Although FEAT-KD uses the specific form given in Eq. 1 for interpretability reasons, we agree it can also be counted as a type of  SR algorithm and be positioned in the broader literature of SR algorithms. Using the results from SRBench, across the datasets and random seeds, FEAT-KD is Pareto-optimal for 60.0% of them with respect to the other SR algorithms. The rate at which the other SR algorithms DSR, GP-GOMEA, Operon, gplearn, AFP, AFP_FE, AIFeynman, FEAT, EPLEX, ITEA, SBP-GP, BSR, MRGP, FFX, are 88.8%, 68.8%, 61.3%, 53.8%, 48.8%, 40.0%, 27.5%, 23.8%, 21.3%, 15%, 11.3%, 8.8%, 5.0%, 2.5%, respectively. In SRBench, Pareto-optimal means that FEAT-KD has the optimal trade-off with prediction performance and equation size, in which there are no other SR algorithms with smaller equation size with better prediction performance. Of note are 3 algorithms which appear more frequently than FEAT-KD on the Pareto-optimal front: DSR, GP-GOMEA and Operon, of which FEAT-KD has a better test_R2 than DSR in 86.3% of the experiments, a better test_R2 than GP-GOMEA in 68.8% of experiments, and smaller size than Operon in 100% of experiments. A Pareto plot will be able to demonstrate this better, which we will include in the revision, but generally speaking, in terms of test_R2, the trend is Operon>FEAT-KD>GP-GOMEA>DSR and in terms of model size, the trend is DSR<GP-GOMEA<FEAT-KD<Operon. Thus, FEAT-KD is frequently Pareto-optimal, providing a new point on the optimal Pareto front with a unique trade-off between test_R2 and model size. We will include these results in Appendix G, “FEAT-KD as an SR Algorithm”.
>
> It is also noteworthy that FEAT-KD appears much more frequently on the Pareto-optimal front than FEAT (60.0% vs 23.8%).
> > Methods And Evaluation
>
> R21: We think yes to a large extent, and the reasons to support this are kindly summarized succinctly from reviewer kPdZ: “the use of the datasets to benchmark the performance is appropriate and consistent with the existing literature on the topic(s), considering that the datasets cover different types - i.e., synthetic datasets, single-target & multi-target regression datasets.” And from reviewer Dqc6: “rigorously evaluates multiple datasets with repeated splits and statistical testing (Wilcoxon tests, Bonferroni correction).”
> > Experimental Designs
>
> R22: We address the datasets and baselines above. As for the ablation study, we thank the reviewer for the suggestion, and have substituted with Operon, PySR, DSR and ParFam. Similar to SR benchmarking, there is a trade-off between model size and prediction performance. The frequency of being on the Pareto-front for FEAT-KD is 88.6%. Whereas substitution with DSR, PySR, ParFam and Operon yielded 54.4%, 25.3%, 19.0% and 1.3% frequency of being Pareto optimal, respectively. We found that DistilSR in FEAT-KD is best able to exploit the expressivity of the search space of short equations to better model the learnt representations by TabNet without overfitting. The results also show the task of fitting the learnt representations is different from normal regression and that SR algorithms that tend to generate shorter expressions perform better. Finally an additional strength of using DistilSR over other algorithms is that the standard deviation over random seeds is the lowest because the equation structure search space is the same for every run. We will include this discussion in Appendix H, “Ablation: Substituting DistilSR”.
> > Other Strengths And Weaknesses
>
> R23: In cases where exhaustive search with too many base functions are expensive, non-exhaustive SR algorithms, such as those used in response R22, can be used. We also intentionally chose a smaller subset of function for FEAT-KD (see Table 3) for interpretability.
> > Comments
>
> R24: We will draw a vertical line to segregate models with FEAT-like structure (i.e., Eq. 1), and deep-learning models.
>
> R25: We thank the reviewer and will include this baseline for multi-target regression. Other than compute, it is also less interpretable because it does not reuse learnt features across targets.
> > Questions
>
> R26: FEAT was allowed to use all base functions in Table 2 in the main text and FEAT-KD was only allowed to use base functions in Table 3 in the main text for increased interpretability.
>
> R27: TabNet itself was run on the same single CPU which has 36 cores and a walltime of 3600 second (or equivalently 129600 core-seconds) counts into the time budget.
>
> R28: We will bold the winner in Table 7.

---

> > ### Comment · Reviewer_9uvT · 2025-04-07
> >
> > Dear authors, thank you for the detailed rebuttal.
> >
> > Most of my concerns have been addressed. One last question regarding R20:
> >
> > Are these results with respect to the extended datasets in PMLB used in FEAT or on the smaller subset?
> >
> > ---
> >
> > Edit 1 : Can you also report the results for R^2,  Model Size, and Training Time, as in Figure 1 in the Srbench paper [1] (as a table here instead of a figure)?  This makes it easier to compare the results.
> >
> > ----
> >
> > Edit 2: Thank you for the additional experiments and metrics! In my opinion, this highlights the contribution made by your approach and should be part of the paper. I have increased my score now. By the way, I believe you mixed up the first two columns.

---

> > > ### Author Response · Authors · 2025-04-08
> > >
> > > > Dear authors, thank you for the detailed rebuttal.
> > > Most of my concerns have been addressed. One last question regarding R20:
> > > Are these results with respect to the extended datasets in PMLB used in FEAT or on the smaller subset?
> > >
> > > We thank the reviewer for the kind words and the opportunity to reply. The results in R20, which are compared against SRBench results, are with respect to the smaller subset. Only the non-SRBench/non-SR results (i.e., R19, R22) are with respect to the extended datasets in PMLB used in FEAT. This is because our main results are on 100 random seeded (random_state_list = [0,1,…, 99]) splits of 20% test set size, whereas the SR algorithms evaluated in SRBench requires 10 specific random seeds (random_state_list = [860, 5390, 6265, 11284, 11964, 15795, 21575, 2218, 23654, 29802]) of 25% test set size, which is a different setting from the settings used in our paper, FEAT-KD, FEAT and TabNet. Hence FEAT-KD had to be trained and computed on their specified settings to be directly compatible with SRBench results. So, at the point of our first reply, R20 is only with respect to the smaller subset.
> > >
> > > With more time provided for computation, we now have SRBench-compatible results for the full extended datasets as well, using the exact same 10 random seeds (random_state_list = [860, 5390, 6265, 11284, 11964, 15795, 21575, 2218, 23654, 29802]) for the 25% test splits that SRBench uses.
> > >
> > > The results give similar conclusions to R20 as shown below (**the differences from R20 are bolded**):
> > >
> > > """
> > > We thank the reviewer for the pointer. Although FEAT-KD uses the specific form given in Eq. 1 for interpretability reasons, we agree it can also be counted as a type of SR algorithm and be positioned in the broader literature of SR algorithms. Using the results from SRBench, across the **extended datasets in PMLB used in FEAT** and random seeds, FEAT-KD is Pareto-optimal for **62.0**% of them with respect to the other SR algorithms. The rate at which the other SR algorithms DSR, GP-GOMEA, Operon, gplearn, AFP, AFP_FE, AIFeynman, FEAT, EPLEX, ITEA, SBP-GP, BSR, MRGP, FFX, are **83.6%, 75.9%, 52.6%, 48.6%, 25.5%, 25.8%, 8.3%, 23.0%, 23.2%, 4.9%, 8.8%, 15.2%, 1.5%, 2.7%**, respectively. In SRBench, Pareto-optimal means that FEAT-KD has the optimal trade-off with prediction performance and equation size, in which there are no other SR algorithms with smaller equation size with better prediction performance. Of note are **only 2** algorithms which appear more frequently than FEAT-KD on the Pareto-optimal front: DSR, GP-GOMEA, of which FEAT-KD has a better test_R2 than DSR in **83.8%** of the experiments and a better test_R2 than GP-GOMEA in **70.7%** of experiments. A Pareto plot will be able to demonstrate this better, which we will include in the revision, but generally speaking **(and including Operon which appears less frequently that FEAT-KD on the Pareto-optimal front but is the 4th most frequent)**, in terms of test_R2, the trend is Operon>FEAT-KD>GP-GOMEA>DSR and in terms of model size, the trend is DSR<GP-GOMEA<FEAT-KD<Operon. Thus, FEAT-KD is frequently Pareto-optimal, providing a new point on the optimal Pareto front with a unique trade-off between test_R2 and model size. We will include these results in Appendix G, “FEAT-KD as an SR Algorithm”.
> > >
> > > It is also noteworthy that FEAT-KD appears much more frequently on the Pareto-optimal front than FEAT (**62.0**% vs **23.0**%).
> > > """
> > >
> > > > Can you also report the results for R^2, Model Size, and Training Time, as in Figure 1 in the Srbench paper [1] (as a table here instead of a figure)? This makes it easier to compare the results.
> > >
> > > We take the “the mean of the median test set performance” as described in the captions on all 88 PMLB datasets used in FEAT. The SRBench GitHub code in $\texttt{postprocessing/blackbox\\_results.ipynb}$ code cell 5 seems to be plotting the median across all datasets and random_state instead. Because of this, to be clear without doubts, we take “the mean of the median test set performance” literally and this is done in Python pandas:
> > >
> > > #First, take the median value across all random_state
> > >
> > > grouped_medians = df.groupby(['algorithm', 'dataset'])[['model_size', 'r2_test', 'training time (s)']].median()
> > >
> > > #Then, take the mean of the median
> > >
> > > result = grouped_medians.groupby('algorithm')[['model_size', 'r2_test', 'training time (s)']].mean()
> > >
> > > |SR Algorithm|model_size|r2_test|training time(s)|
> > > |-|-|-|-|
> > > |AFP|35.4|0.688|2790|
> > > |AFP_FE|36.0|0.700|2850|
> > > |AIFeynman|2240|-4.03|82900|
> > > |BSR|19.8|0.245|13100|
> > > |DSR|8.88|0.598|35100|
> > > |EPLEX|56.3|0.814|7530|
> > > |FEAT|79.1|0.847|6750|
> > > |FEAT-KD|49.0|0.851|1550|
> > > |FFX|1630|0.00612|210|
> > > |GP-GOMEA|25.1|0.794|7610|
> > > |ITEA|107|0.663|6240|
> > > |MRGP|10800|0.521|201000|
> > > |Operon|63.7|0.853|1350|
> > > |SBP-GP|693|0.851|155000|
> > > |gplearn|17.6|0.563|23300|
> > >
> > > ,which we will add in the main text.
> > >
> > > We hope that these address the reviewer’s remaining concern and the reviewer would consider recommending the acceptance of this paper.

---

### Official Review · Reviewer_Dqc6 · 2025-03-12

**Overall Recommendation:** 3

**Summary:**

The paper presents FEAT-KD, a method that distills knowledge from a TabNet deep neural network into interpretable symbolic regression models. It replaces traditional genetic programming (GP) used in FEAT with a neural-guided symbolic regression approach, producing concise mathematical expressions. Key contributions include demonstrating competitive performance compared to TabNet, significantly simpler symbolic models, and support for multi-target regression tasks.

**Claims And Evidence:**

Claims of competitive accuracy, improved interpretability, and successful multi-target support are supported by empirical results (15 single-target and 5 multi-target datasets). Statistical analyses confirm significant improvements over original FEAT variants.

**Essential References Not Discussed:**

The major benefit of this work is a form of interpretable machine learning for tabular data. Hence, InterpreTabNet (https://arxiv.org/abs/2406.00426) from ICML 2024 should be very relevant to this paper.

**Experimental Designs Or Analyses:**

The paper rigorously evaluates multiple datasets with repeated splits and statistical testing (Wilcoxon tests, Bonferroni correction). A limitation: evaluation on very high-dimensional or extremely large-scale data is not addressed. Additionally, FEAT-KD generates symbolic expressions as linear combinations of numeric features, which naturally aligns with continuous-valued predictions thus, scope of this method is limited to only regression tasks which is not ideal.

**Methods And Evaluation Criteria:**

TabNet's learned feature representations guide symbolic regression, greatly reducing the search space. Evaluation criteria are appropriate.

**Other Comments Or Suggestions:**

N/A

**Other Strengths And Weaknesses:**

**Originality**

Work is novel. As far as I know, I have not come across the use of symbolic regression within interpretable tabular learning.

**Significance**

Good significance. Addresses the critical need for interpretable ML models without substantial performance loss.

**Clarity**

Good. Although Figure 1 seems to be way too large filling 1 whole page and abstract is way too long. Per the guidelines, abstract should be 4-6 sentence long.

**Questions For Authors:**

As mentioned above, I understand that this work is a form of interpretable machine learning for tabular data. However, the interpretability aspect is still unclear to me. There are many nice experiments highlighting your quantitative performance over other symbolic methods as well as remaining somewhat competitive to TabNet barring in mind the interpretability vs. accuracy tradeoff. But I am unable to observe extensive analysis or examples into how in “interpretability” of FEAT can assist practitioners.

I would highly recommend looking into [1] InterpreTabNet to rephrase how the “interpretability” of your method outweighs the sacrifice in accuracy.

[1] Si, J., Cheng, W.Y., Cooper, M. and Krishnan, R.G., 2024. InterpreTabNet: distilling predictive signals from tabular data by salient feature interpretation. *arXiv preprint arXiv:2406.00426*.

**Relation To Broader Scientific Literature:**

Connects with symbolic regression (FEAT, DistilSR, DSR), knowledge distillation, interpretability literature, and multi-target regression research.

**Theoretical Claims:**

No complex theoretical proofs in the paper.

---

> ### Author Rebuttal · Authors · 2025-04-01
>
> > General
>
> Please see response R1 to reviewer mwAW.
>
> > Experimental Designs Or Analyses
>
> R14: We thank the reviewer for the suggestion, the original FEAT is designed for regression tasks [1] and since a large amount of evaluation was required to verify both single and multi-target regression, we did not focus on classification in this work and instead focused on existing datasets that TabNet and FEAT evaluates on. However, the reviewer is right FEAT-KD can be adapted for classification by changing using logistic regression in Phase 5, which we show is possible via the Diabetes dataset in the revision Appendix I, “Extension to Classification”, but the thorough evaluation and design would be left to future work since we would like the space to focus more thoroughly on single and multi-target regression tasks.
>
> [1] La Cava, W., et al. (2019). Learning concise representations for regression by evolving networks of trees.
>
> > Essential References Not Discussed
>
> R15: We thank the reviewer for the pointer to InterpreTabNet. InterpreTabNet has 2 main contributions, both of which are relevant and we will address: i). A regularization scheme that maximizes diversity between masks in the TabNet architecture, ii). capturing feature interdependencies by prompting LLMs with the learnt masks.
>
> For i)., InterpreTabNet works very similarly to TabNet, just with extra regularization, so we could easily apply FEAT-KD techniques by replacing TabNet with InterpreTabNet in the code implementation. The results yield stronger performance on some datasets, but not on all, consistent with the results in [2] itself. Thus, InterpreTabNet allows for a variant of FEAT-KD which practitioners should try as it is dataset dependent (depends on whether reuse of features across attention masks in TabNet is desirable for the problem). We will include these results in Appendix J, “InterpreTabNet Variant of FEAT-KD” and show a subset of test $R^2$ below:
>
> |Dataset|FEAT-KD (TabNet)|FEAT-KD (InterpreTabNet)|
> |-|-|-|
> |bodyfat|0.987 (0.0058)|0.990 (0.0051)|
> |cpu_act_197|0.976 (0.0014)|0.977 (0.0015)|
> |cpu_act_573|0.976 (0.0014)|0.976 (0.0016)|
> |cpu_small|0.968 (0.0015)| 0.966 (0.0015)|
> |house_8L|0.585 (0.01)|0.588 (0.0096)|
> |houses|0.664 (0.0078)|0.660 (0.0053)|
> |pm10|0.181 (0.026)|0.196 (0.042)|
> |puma8NH|0.618 (0.026)|0.596 (0.0086)|
>
> For ii). InterpreTabNet uses LLMs to generate linguistic interpretations of the masks obtained which is not mutually exclusive with FEAT-KD. In fact, InterpreTabNet and FEAT-KD complement each other, in which InterpreTabNet generates a simplified qualitative description, whereas FEAT-KD generates a simplified quantitative description, which we thank the reviewer for highlighting. Thus, from the trained mask, 2 interpretations can be generated, one from using an LLM to generate a linguistic interpretation and the other from the learnt equation from FEAT-KD. We will include these results in Appendix J, “InterpreTabNet Variant of FEAT-KD”.
>
> Thus, the strengths of InterpreTabNet can be subsumed into FEAT-KD by replacing TabNet with InterpreTabNet, which we thank the reviewer for the recommendation and have included in the revision.
>
> [2] Si, J. Y. H., e al. InterpreTabNet: Distilling Predictive Signals from Tabular Data by Salient Feature Interpretation.
>
> > Other Strengths And Weaknesses
>
> R16: For clarity, we thank the reviewer for the suggestion and will reduce the size of Figure 1 by 70% and reduce the number of sentences in the abstract.
>
> > Questions For Authors
>
> R17: For interpretability aspect question, the practical interpretability is argued to be measured in proxy by model size in [3,4] and the simplicity of the symbolic expressions has been argued to be interpretable via disentanglement [5] and also validated by clinicians in studies which uses FEAT structure (i.e., Eq. 1) [6]. We will include this discussion in Section 4.
>
> [3] Lage, I., et al. (2019). An evaluation of the human-interpretability of explanation.
>
> [4] Abdul, A., et al. (2020). COGAM: measuring and moderating cognitive load in machine learning model explanations.
>
> [5] La Cava, W., et al. (2019). Learning concise representations for regression by evolving networks of trees.
>
> [6] La Cava, W., et al. (2023). A flexible symbolic regression method for constructing interpretable clinical prediction models.
>
> R18: We thank the reviewer for the suggestion and will tackle “interpretability” via a qualitative and quantitative approach, where we will make clear that we can leverage InterpreTabNet for qualitative interpretability via capturing feature interdependencies by prompting LLMs with the learnt masks and we can assess quantitative interpretability via computable metrics such as model size. We will also tackle the sacrifice and tradeoff via Pareto fronts, which we address in response R20 and R22 as well.

---

> > ### Comment · Reviewer_Dqc6 · 2025-04-02
> >
> > Dear Authors,
> >
> > Thank you for the thorough rebuttal, the clarification and experiments on InterpreTabNet are greatly appreciated and helpful. I would really like to raise my score but it's just that it currently lacks classification task experiments. If there is any possibility in which you can include them it will make your paper really impactful.

---

> > > ### Author Response · Authors · 2025-04-03
> > >
> > > > Dear Authors, Thank you for the thorough rebuttal, the clarification and experiments on InterpreTabNet are greatly appreciated and helpful. I would really like to raise my score but it's just that it currently lacks classification task experiments. If there is any possibility in which you can include them it will make your paper really impactful.
> > >
> > > We thank the reviewer for the acknowledgement, suggestions and the opportunity to reply. For the remaining concern on classification task experiments, one possible way FEAT-KD can be adapted for classification is by using logistic regression in ‘Phase 5’ instead of linear regression. Rather than using
> > >
> > > $\hat{y}(\mathbf{x}) = \phi(\mathbf{x})^T \hat{\beta}$,
> > >
> > > as shown in Eq. 1 for regression, we use a logistic regression model for classification. That is, for an input $\mathbf{x}$ and $K$ classes, the model is defined as
> > >
> > > $P(y=k \mid \mathbf{x}) = {\exp(\phi(\mathbf{x})^T \hat{\beta}_k)}/z$
> > >
> > > where  $z={\sum_{j=1}^{K} \exp(\phi(\mathbf{x})^T \hat{\beta}_j)}$, $\phi(\mathbf{x})$ is the feature vector, and $\hat{\beta}_k$ is the coefficient vector for class $k$.
> > >
> > > The predicted class is given by
> > >
> > > $$
> > > \hat{y} = \arg\max_k \; P(y=k \mid \mathbf{x}).
> > > $$
> > >
> > > We denote the methods with the prefix ‘c’ to differentiate them from the regression case. To evaluate the classification performance, we replaced $R^2$ score with both $accuracy$ and $F1$ score instead. The datasets chosen are from the **same sources as related literature in classification**  [1,2], and repeated across 100 random seeded data splits.
> > >
> > > Below are the truncated accuracy of some experiments due to character limits (truncated to 3 s.f. with standard deviation in brackets):
> > >
> > > |Dataset|cFEAT-KD (TabNet)|cFEAT-KD (InterpreTabNet)|cFEAT|cFEAT-Corr|cFEAT-CN|TabNet|InterpreTabNet|
> > > |-|-|-|-|-|-|-|-|
> > > |chess|0.969 (0.0033)|0.969 (0.0034)|0.949 (0.0055)|0.943 (0.0042)|0.945 (0.0036)|0.985 (0.00086)|0.987 (0.00086)|
> > > |ionosphere|0.914 (0.014)|0.909 (0.011)|0.885 (0.011)|0.868 (0.016)|0.876 (0.012)|0.873 (0.0061)|0.871 (0.0057)|
> > > |sonar|0.815(0.025)|0.806 (0.035)|0.705 (0.012)|0.702 (0.030)|0.746 (0.017)|0.635 (0.019)|0.682 (0.036)|
> > > |spambase|0.934 (0.0024)|0.935 (0.0024)|0.898 (0.0091)|0.895 (0.0039)|0.901 (0.0065)|0.934 (0.0016)|0.934 (0.0019)|
> > > |spectf|0.843 (0.018)|0.845 (0.019)|0.812 (0.021)|0.776 (0.011)|0.793 (0.017)|0.836 (0.022)|0.812 (0.025)|
> > > |tokyo1|0.918 (0.0030)|0.920 (0.0073)|0.907 (0.012)|0.906 (0.0033)|0.906 (0.0047)|0.908 (0.010)|0.914 (0.0052)|
> > > |Diabetes|0.765 (0.0099)|0.770 (0.014)|0.738 (0.0079)|0.733 (0.0073)|0.737 (0.015)|0.775 (0.015)|0.777 (0.012)|
> > > |Forest Cover Type|0.755 (0.0017)|0.753 (0.0017)|0.748 (0.0021)|0.748 (0.0012)|0.749 (0.0010)|0.753 (0.0022)|0.754 (0.0022)|
> > > |…|
> > > |...|
> > >
> > > And below are the corresponding $F1$-scores:
> > >
> > > |Dataset|cFEAT-KD (TabNet)|cFEAT-KD (InterpreTabNet)|cFEAT|cFEAT-Corr|cFEAT-CN|TabNet|InterpreTabNet|
> > > |-|-|-|-|-|-|-|-|
> > > |chess|0.970 (0.0041)|0.969 (0.0034)|0.952 (0.0051)|0.947 (0.0043)|0.948 (0.0039)|0.986 (0.00092)|0.987 (0.0015)|
> > > |ionosphere|0.912 (0.015)|0.908 (0.011)|0.905 (0.011)|0.898 (0.016)|0.894 (0.015)|0.905 (0.0052)|0.902 (0.0035)|
> > > |sonar|0.814 (0.025)|0.805 (0.035)|0.678 (0.036)|0.702 (0.040)|0.721 (0.049)|0.591 (0.027)|0.623 (0.040)|
> > > |spambase|0.934 (0.0025)|0.935 (0.0025)|0.865 (0.012)|0.866 (0.0059)|0.869 (0.0092)|0.918 (0.0031)|0.916 (0.0035)|
> > > |spectf|0.836 (0.024)|0.844 (0.021)|0.863 (0.019)|0.850 (0.013)|0.856 (0.017)|0.887 (0.021)|0.876 (0.020)|
> > > |tokyo1|0.919 (0.0054)|0.920 (0.0073)|0.929 (0.0095)|0.926 (0.0066)|0.927 (0.0054)|0.930 (0.0085)|0.935 (0.0080)|
> > > |Diabetes|0.728 (0.012)|0.734 (0.017)|0.699 (0.0088)|0.693 (0.0094)|0.697 (0.016)|0.740 (0.019)|0.742 (0.014)|
> > > |Forest Cover Type|0.554 (0.0041)|0.550 (0.0039)|0.538 (0.0069)|0.536 (0.0045)|0.539 (0.0043)|0.552 (0.0052)|0.554 (0.0058)|
> > > |...|
> > > |...|
> > >
> > > Tables for p-values and model sizes are also available. Similar to regression results, cFEAT-KD is smaller in model size than cFEAT, and both cFEAT-KD and cFEAT are much smaller than TabNet. Likewise, cFEAT-KD has statistically significant outperformance on most datasets compared to all variants of cFEAT. And finally, like regression, although cFEAT-KD cannot consistently outperform TabNet (which is to be expected due to being significantly less complex), it still performs competitively and on slightly more than a third of the datasets, it even has better performance on the test data in terms of $accuracy$ and $F1$ score due to its regularization effect. As a continuation from response R14, these will be included in the revision in the main text and with raw results Tables in Appendix I, “Extension to Classification”.
> > >
> > > We hope that these address the reviewer’s remaining concern and the reviewer would consider recommending the acceptance of this paper.
> > >
> > > [1] Arik, S. Ö., et al. (2021). TabNet: Attentive interpretable tabular learning.
> > >
> > > [2] La Cava, W., et al. (2023). A flexible symbolic regression method for constructing interpretable clinical prediction models.

---

### Official Review · Reviewer_kPdZ · 2025-03-25

**Overall Recommendation:** 4

**Summary:**

The FEAT framework typically uses genetic algorithms to derive the concise-feature representations. The linear combination of these concise-representation features is then the predicted output of this model. TabNet is a deep-learning based model for tabular data that can be used for single/multi-target regression. In this work, the authors propose a novel framework called FEAT-KD wherein they combine the ideas of TabNet and FEAT along with knowledge distillation and an exhaustive-search symbolic regression instead of genetic algorithms to identify the concise-representations. More specifically, the authors combine the “steps” from the TabNet architecture, and then apply symbolic regression on the output of each “step” along with the extracted TabNet masks for feature selection to distill it into a concise mathematical equation. Finally, they perform a simple linear regression on the concise-representations to end up with a “white-box” model structure that is comparable to FEAT in terms of interpretability but without using genetic programming.

The main results show that FEAT-KD performs better than FEAT and other variants such as FEAT-Corr, and FEAT-CN across a range of single-target regression and multi-target regression datasets while still being similar to the TabNet (large sized teacher model) model(s). The authors also show that the resulting model size (in terms of the total number of functions and terminals in the model) is on-average the smallest across most of the model-dataset combinations evaluated upon, and they show that these improvements in model size/performance are statistically significant by using the one-sided Wilcoxon signed-rank test. Thus, the authors show that we can achieve the best of both worlds - the improvements shown by TabNet along with the interpretability of the FEAT framework, for tabular data.

**Claims And Evidence:**

The claims made by the authors are clearly discussed.
Claims made -
1. FEAT-KD results is close to the performance of the TabNet model while improving the interpretability from being a black-box model to a white-box model.
2. FEAT-KD has the best average prediction performance that is consistent and statistically significant among symbolic models evaluated.
3. FEAT-KD has the lowest average model size among all models evaluated and has statistically significant evidence that it is consistently smaller.
4. FEAT-KD easily supports multi-target regression.
5. FEAT-KD has some regularization effect over TabNet.

In general, the claims are well-substantiated with numerical comparisons and p-values for statistical significance of the results.

**Essential References Not Discussed:**

The following work(s) are not essential references required to understand the content of the paper, but they seem to be relevant to the ideas proposed here and the authors would be justified to have mentioned them in the related works section or done some comparisons against these methods, especially methods that also use symbolic regression on top of deep-learning based methods to find interpretable equations.
Cramer et al - Discovering Symbolic Models from Deep Learning with Inductive Biases (NeurIPS 2020)

**Experimental Designs Or Analyses:**

Yes. Experimental design. An issue with the current setup is that the authors only considered other symbolic methods and the teacher model (TabNet) for comparisons but do not consider other similar approaches that use symbolic regression on top of deep-learning methods. Considering these baselines too would have made the evaluation section stronger that it currently is.

**Methods And Evaluation Criteria:**

The proposed methods of using the TabNet architecture to distill the “step” outputs along with the masks to learn concise features with symbolic regression via an exhaustive search makes sense for the problem of tabular regression. Further, the use of the datasets to benchmark the performance is appropriate and consistent with the existing literature on the topic(s), considering that the datasets cover different types - i.e., synthetic datasets, single-target & multi-target regression datasets. The use of the evaluation metrics - R2 score, model size (in terms of functions and terminals) is also appropriate and relevant for the field.

**Other Comments Or Suggestions:**

N/A

**Other Strengths And Weaknesses:**

Strengths:
1. The idea of combining existing methods such as TabNet and FEAT using symbolic regression with knowledge distillation instead of genetic programming to formulate FEAT-KD is novel.
2. The straightforward extension from single-target regression to multi-target regression will have a big impact on critical industries such as healthcare.
3. The resulting method ends up making the TabNet architecture “more interpretable”.

Weakness:
1. Even though the experiments and ablations presented are comprehensive they’d have been stronger by including comparisons against methods that also use symbolic regression methods (with genetic programming) along with Deep-Learning techniques to derive interpretable equations such as the use of symbolic regression on top of GNN networks.

**Questions For Authors:**

1. In the evaluations section, is there a reason why other similar approaches as the use of symbolic regression on top of Graph Neural networks are not considered as another baseline?
2. In terms of interpretability, how would the existing interpretable methods be applied on TabNet and fare vs the interpretable FEAT-KD in terms of interpretability? Ref - Fig.2 from Borisov et al., Deep Neural Networks and Tabular Data: A Survey

**Relation To Broader Scientific Literature:**

1. Symbolic regression methods use genetic programming to identify the concise features. Feat-KD replaces this step by finding a weighted linear combination of concisely represented symbolic features via piece-wise knowledge distillation corresponding to each “step” of the trained TabNet model.

2. Feat-KD is built on top of existing methods such as TabNet and FEAT. While TabNet is a deep-learning-based model and inherently a “black-box” model, FEAT, on the other hand, is a “white-box” algorithm. Feat-KD achieves competitive performance with TabNet on single and multi-target regression datasets while also making the resulting model more interpretable compared to the original TabNet.

3. Across all the eval datasets considered Feat-KD consistently outperforms other symbolic methods such as FEAT and variants of FEAT. The authors show that FEAT-KD consistently outperforms other symbolic models such as FEAT and its variants both in performance (R2 score) and also in terms of model size (smaller size) compared to other symbolic methods.

**Theoretical Claims:**

The paper does not provide any proofs for theoretical claims, the claims made are empirical in nature and they are validated experimentally.

---

> ### Author Rebuttal · Authors · 2025-04-01
>
> > General
>
> We thank the reviewer for the thorough review and suggestions. We address the comments and implement the suggestions in our individual responses to the reviewers. For convenient referencing to a response, we use the notation R{number} to label a response. We hope that if the responses address the reviewer’s concerns, the reviewer would consider increasing the score.
>
> > Experimental Designs Or Analyses
>
> R9: We thank the reviewer for the suggestion and have included other comparisons with DSR and AIFeynman, methods that uses some elements of deep-learning to find interpretable equations, which we will include in Appendix G & H (please see responses R20 & R22).
>
> > Essential References Not Discussed
>
> R10: We thank the reviewer for the pointer, we will reference the influential work by Cranmer, M., et al. and discuss the differences. For evaluation purposes the method by Cranmer, M., et al. is designed for “interacting particle” systems, where the problem is modelled as individual particles, each with their own dataset, and the task is to find the interactions between them. It is thus not adaptable to common tabular regression done by FEAT and TabNet. We have in the revision, however, compared to DSR and AIFeynman, which uses elements of deep-learning to find interpretable equations, which we will include in Appendix G & H (please see responses R20 & R22). DSR and AIFeynman work very differently from FEAT-KD. While FEAT-KD uses deep-learning to model the distribution of the data directly and distill components piecewise, DSR uses deep-learning for modelling the distribution of candidate equations and AIFeynman uses a neural network fitting to remove noise from data. Another difference is the objectives of the algorithms are different, FEAT-KD tries to find a structure in the form of Eq. 1, which has been validated by clinicians to be interpretable, whereas DSR and AIFeynman uses different structures. We will add this discussion in the main text Section 4.
>
> > Other Strengths And Weaknesses
>
> R11: We thank the reviewer and will add more comparisons (we respond to this above in R9 & R10, which is in turn responded partially with responses R20 & R22).
>
> > Questions For Authors
>
> R12: For Q1, the use of symbolic regression on top of Graph Neural networks is not adaptable for tabular regression which TabNet and FEAT addresses. However, we recognize that although FEAT-KD uses the specific form given in Eq. 1 for interpretability reasons, it can also be counted as a type of SR algorithm and be positioned in the broader literature of SR algorithms. We respond to this above in R9 & R10, which is in turn responded partially with responses R20 & R22.
>
> R13: For Q2, we thank the reviewer for the reference. The current interpretable methods applied on TabNet look at feature attributions, similar to the trained masks in TabNet. FEAT-KD retains this exact property and these current methods are still applicable to FEAT-KD. However, FEAT-KD takes this one step further by finding a concise faithful equation that captures the behaviour of the learnt representation. Thus, in terms of interpretability, FEAT-KD subsumes the interpretability that TabNet possesses and further adds an additional aspect of interpretability via the learnt equations.

---

### Official Review · Reviewer_mwAW · 2025-03-25

**Overall Recommendation:** 3

**Summary:**

This paper introduces FEAT-KD, a method that transfers the strengths of TabNet and FEAT to create concise, interpretable models for both single-target and multi-target regression. The method distills pieces of a trained TabNet into short symbolic expressions using an exhaustive search algorithm (DistilSR). These symbolic expressions are then combined into a weighted linear model. Extensive experiments on various datasets show that FEAT-KD achieves competitive predictive performance with significantly smaller model sizes compared to both TabNet and previous FEAT variants.

**Claims And Evidence:**

Yes, most of them are supported.

**Essential References Not Discussed:**

No

**Experimental Designs Or Analyses:**

Yes, I think they are valid.

**Methods And Evaluation Criteria:**

Yes

**Other Comments Or Suggestions:**

See the questions for authors.

**Other Strengths And Weaknesses:**

Strengths:
1. The proposed method effectively converts a deep learning model into a white-box model by producing symbolic, concise representations.
2. FEAT-KD supports multi-target regression, which is not available in the original FEAT and its variants.
3. The paper provides an extensive experimental evaluation on a variety of datasets, with statistical tests (e.g. Bonferroni-adjusted p-values) to support its claims.
4. The five-phase approach is clearly laid out, showing how deep-learning based feature selection and symbolic regression are integrated (Sec 3).

Weaknesses:
1. The use of an exhaustive search algorithm (DistilSR) for symbolic regression may not scale well if more complex or higher-dimensional feature subsets are required.
2. The paper does not fully address the sensitivity of the method to hyperparameters such as the number of features selected from the masks (top 3) or the choices for Nd and Nsteps. The current ablation studies (Tables 9–11) are informative but could be discussed more in the main text. (But in general, there are a bit too many tables in the main text...)
3. While the paper claims improved interpretability via model size reduction, there is little discussion on how the simplicity of the symbolic expressions translates to practical interpretability for end users?
4. Some methodological detailsare not fully detailed, esp on the feature selection using TabNet masks (Phase 3) and the computational cost of the exhaustive search.

**Questions For Authors:**

1. Please provide more details on the computational cost and scalability of the DistilSR step. A brief discussion or additional experimental data in sec 3 would be helpful.
2. Clarify the exact procedure for selecting the top 3 features from the TabNet masks. Including a more detailed example or additional explanation near Section 3 could improve clarity.
3. Expand the discussion on the sensitivity of the method to the hyperparameters Nd and Nsteps. It would be beneficial to integrate a short ablation study discussion from Tables 9–11 into the main text.
4. Explain a bit further how the AFI-MSE metric (sec 3) compares to traditional MSE, perhaps with a simple illustrative example.

**Relation To Broader Scientific Literature:**

Good

**Theoretical Claims:**

NA, there is not theory

---

> ### Author Rebuttal · Authors · 2025-04-01
>
> > General
>
> R1: We thank the reviewers for the thorough reviews and suggestions. We address the comments and implement the suggestions in our individual responses to the reviewers. For convenient referencing to a response, we use the notation R{number} to label a response (e.g., this response is labelled as R1). We hope that if the responses address the reviewer’s concerns, the reviewer would consider increasing the score.
>
> > Other Strengths And Weaknesses
>
> R2: For weakness 1, given that the premise in which a user would want to obtain a FEAT-like model (i.e., Eq. 1) is to obtain interpretability, we do not foresee having a large feature subset because that would lead to a long expression and is hence no longer interpretable [1,2]. If the feature subsets are high-dimensional, then the initial benefit of choosing a FEAT-like structure rather than a black-box model is largely lost.
>
> Additionally, in Reply R22 we show that FEAT-KD can use alternative SR algorithms that scale better. However, note their prediction performance is at the expense of loss of interpretability (measured by increased size) or may even predict worse. DistilSR exploits the high expressivity of the search space of short expressions instead of greedily searching longer expressions at the expense of interpretability.
>
> R3: For weakness 2, we thank the reviewer for the suggestion and will make changes to the organization of the paper. Currently, we describe in Appendix D how we obtained the hyperparameter values via preliminary experiments on a small subset of the data to prevent data leakage, e.g., target 1 of atp1d. For the selection of “top 3”, it was not tuned but rather a decision for interpretability which is consistent with other TabNet literature [3]. We will move Tables 6 & 7 to the Appendix and split Table 8 into 2 parts (one in the main text and the other half in the Appendix) in order to move the current discussion of hyperparameters in Appendix D to the main text.
>
> R4: For weakness 3, the practical interpretability is argued to be measured in proxy by model size in [1,2] and the simplicity of the symbolic expressions has been argued to be interpretable via disentanglement [4] and also validated by clinicians in studies which uses FEAT structure (i.e., Eq. 1) [5]. We will include this discussion in Section 4.
>
> R5: For weakness 4, we will clarify that in Phase 3, we will extract the multiple mask matrices learnt by TabNet. For each mask matrix, we average the mask across the datapoints to obtain a single averaged value per feature. The 3 features with the highest averaged value are deemed to be the least unmasked and will be the features that are used in building the equation obtained in Phase 4. The computational cost of the exhaustive search is approximately O($d^l$), where $d=9$ is the number of different terminals and symbols, and $l=5$ is the length of the expression. We will be clearer on these in the main text Section 3.
>
> [1] Lage, I., et al. (2019). An evaluation of the human-interpretability of explanation.
>
> [2] Abdul, A., et al. (2020). COGAM: measuring and moderating cognitive load in machine learning model explanations.
>
> [3] Si, J. Y. H., e al. InterpreTabNet: Distilling Predictive Signals from Tabular Data by Salient Feature Interpretation.
>
> [4] La Cava, W., et al. (2019). Learning concise representations for regression by evolving networks of trees.
>
> [5] La Cava, W., et al. (2023). A flexible symbolic regression method for constructing interpretable clinical prediction models.
>
> > Questions For Authors
>
> R6: For Q1 and Q2, please see response R5. In addition, response R20, where we use other SR algorithms may interest the reviewer as well.
>
> R7: For Q3, we thank the reviewer for the suggestion and will include these in the main text per the reorganization of tables outlined in response R3.
>
> R8: For Q4, consider a simple example with true outputs $y=[0,12,28]$ and inputs $x_1=[1,2,3]$ and $x_2=[4,5,6]$. A candidate equation $x_1 \times x_2$, produces predictions $[4,10,18]$, which under traditional MSE yield a relatively high value that would lead to the equation not being picked. However, when using AFI-MSE, which first optimally scales and shifts the predictions to best match the true values, we can find parameters $a$ and $b$ (specifically, $a=−8, b=2$) such that the adjusted prediction $-8 + 2 \times (x_1 \times x_2)$ exactly equals $y$, resulting in a zero error. Thus, AFI-MSE recognizes that while the candidate's predictions differ in scale and shifts, their underlying pattern aligns perfectly with the true values, highlighting the candidate's potential as a good fit despite the initial magnitude mismatch. This is suitable particularly for FEAT-KD because in “Phase 5”, this equation will be scaled and shifted anyway because it is being used as a feature in linear regression. This also effectively simplifies the search space. We will include this discussion in the main text Section 3 under AFI-MSE.

---

### Decision · Program_Chairs · 2025-05-01

**Decision:**

Accept (poster)

**Comment:**

Four reviewers submitted reviews for this paper. After the initial reviews, two of the reviewers updated their ratings.

During the reviewer-AC discussion period, all the reviewers shared their thoughts on this paper, considering the author rebuttal.
Through the internal discussions, one of the reviewers changed their position to leaning towards a reject as they consider applying symbolic regression to tabular learning is not novel. A few reviewers are satisfied with the author rebuttal and additional analysis.

It is an interesting study that applies an idea of knowledge distillation to TabNet with some perspective of symbolic regression (SR). Multi-target regression on top of that makes this study a little bit more unique. It may be worth discussing this study with people from the SR community.

Since some reviewers updated their ratings based on the author rebuttal and the amount of additional contents in their rebuttal is significant, I expect the authors to reflect the changes and additional results to the next revision. Specifically, I find the additional experimental results and analysis not detailed enough e.g., hyperparameter tuning, exactly what datasets they used.

As one of the reviewers is now leaning towards rejection and some of the updated ratings are based on the rebuttal, I am concerned about the quality and soundness of the additional contents. While I suggest accepting this work, I would not fight for accepting this work if there is not room in the program.